# NIC-RobustBench: A Comprehensive Open-Source Toolkit for Neural Image Compression and Robustness Analysis

## Abstract

Adversarial robustness of neural networks is a rapidly developing and important research area, ensuring the reliable use of neural network models in areas such as computer vision, natural language processing, and others. With the emergence of neural image compression (NIC) methods and the introduction of the first standard, JPEG AI, the question of robustness has also become critical in image compression. Unstable NIC models are more prone to producing distortions and can be exploited to compromise downstream vision tasks that rely on compressed data. Most existing benchmarks for NIC focus primarily on compression efficiency and reconstruction quality, while prior robustness studies have been limited to a small number of codecs and attacks. To address this gap, we introduce **NIC-RobustBench**, the first open-source framework for evaluating the adversarial robustness of NIC methods. It integrates 6 adversarial attacks and 7 defense strategies in addition to traditional Rate-Distortion (RD) evaluation. The framework includes the largest number of codec types among all known NIC libraries and is easily scalable. The paper provides a comprehensive overview of NIC-RobustBench alongside an extensive robustness evaluation of modern NIC methods. Our code is publicly available at *link hidden for blind review*. We believe our framework will become an essential tool for developing robust neural image compression techniques.

## 1 Introduction

Image compression reduces the number of bits required to represent an image while preserving visual fidelity as much as possible. This task is fundamental in digital media, enabling efficient storage and transmission of the vast amounts of visual data generated in modern applications. With the proliferation of high-resolution imagery and bandwidth-intensive content, effective image compression remains crucial for reducing data storage requirements and network load. Traditionally, image codecs rely on hand-crafted transforms and heuristics; however, the success of deep learning in various computer vision tasks has sparked interest in neural-network-based compression methods (Yang et al. (2020; 2021); Gao et al. (2021)). In recent years, researchers have proposed numerous neural image compression (NIC) approaches that use learned encoders and decoders to replace or augment components of the conventional compression pipeline. These NIC methods have achieved state-of-the-art performance, leading to the establishment of JPEG AI (Ascenso et al. (2023)) — the first image compression standard built entirely upon neural networks.

However, a significant challenge for neural networks (including NIC models) is their vulnerability to adversarial attacks. An adversarial attack involves adding a carefully crafted perturbation to an input that causes a model to produce incorrect or highly degraded outputs (Dong et al. (2018); Chakraborty et al. (2021)). As a result, adversarial robustness has become a rapidly growing area of research in recent years (Zhang et al. (2021); Wei et al. (2024)). Numerous studies have explored adversarial attacks, defenses, and robustness evaluations for various vision tasks — including image classification (Croce et al. (2020)), object detection (Nezami et al. (2021)), and image quality assessment (Antsiferova et al. (2024)) — underscoring the broad importance of this problem. Like other neural vision models, NIC models are not exempt from these vulnerabilities, and they can suffer severe reconstruction errors when subjected to adversarially crafted inputs. For example, as

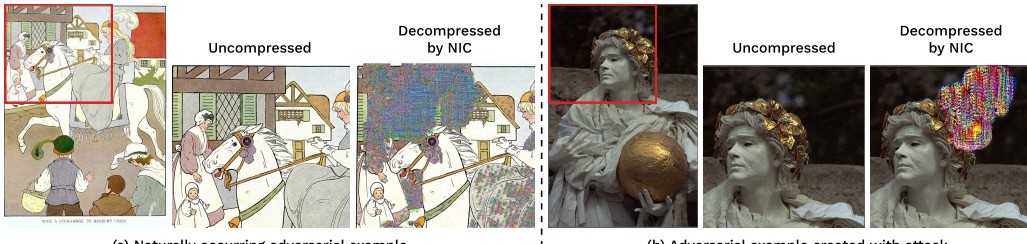

(a) Naturally occurring adversarial example  (b) Adversarial example created with attack

Figure 1: Examples of natural and artificial adversarial examples for neural compression. The left image is taken from Open Images dataset Kuznetsova et al. (2020), the right is taken from Kodak dataset Kodak (1991).

demonstrated in Fig. 2 (a), adding a subtle adversarial perturbation to an input image can significantly reduce the reconstruction quality of a learned codec, despite the perturbation being nearly invisible prior to compression.

There are several reasons why assessing and improving the adversarial robustness of NIC methods is particularly important. First, even in the absence of any malicious interference, unstable behavior in learned compression models can lead to serious visual distortions on certain natural images, as demonstrated in Fig. 1. For instance, Tsereh et al. (2024) recently showed that a state-of-the-art neural codec (JPEG AI) can unexpectedly produce unpleasant artifacts — distortions that do not appear when compressing the same content with a traditional codec at comparable bitrates. Some of these failure cases resemble adversarial examples, pointing to an intrinsic fragility in current NIC approaches. Second, in many real-world scenarios, image compression is applied as a preprocessing step before other computer vision algorithms (e.g., a compressed image is later fed into detection or recognition models). This means an adversary could target the NIC stage as an indirect way to compromise an entire processing pipeline: by introducing perturbations that cause the compressor to output a corrupted image, the attacker can effectively render downstream models (which rely on the compressed output) useless. Therefore, evaluating and enhancing the adversarial robustness of NIC systems is not only critical for the reliability of the compression process itself but also for safeguarding any subsequent tasks that depend on compressed images.

These considerations highlight the need for dedicated tools and methodologies to evaluate and strengthen the robustness of neural compression models. While several studies have investigated the robustness of individual NIC models and proposed specific adversarial attacks, none have consolidated these efforts into a unified open-source library for evaluating neural codec robustness. To this date, all known open-source libraries for image compression focus on optimizing compression performance and neglect the issue of adversarial robustness. To bridge this gap, we propose **NIC-RobustBench** — the first open-source framework for benchmarking and improving the adversarial robustness of NIC models. NIC-RobustBench enables researchers to test a broad range of NIC algorithms under diverse adversarial attack settings and to assess the effectiveness of various defense strategies. Our toolkit is implemented in PyTorch (a widely used deep learning framework for NIC research) to maximize compatibility with existing models, and it is designed to be easily extensible so that new compression codecs can be integrated seamlessly. Furthermore, based on our framework, we establish a robustness benchmark for NIC models that covers a diverse set of codecs, attacks, and defense mechanisms. We hope that NIC-RobustBench will facilitate systematic robustness evaluation in the NIC community and drive the development of more robust neural compression techniques.

Our main contributions are summarized as follows:

- **Open-source benchmark**: We present NIC-RobustBench, a novel open-source library and a corresponding benchmark for evaluating the adversarial robustness of neural image compression methods. The framework supports multiple types of adversarial attacks and defenses for the image compression task, tests attacks impact on downstream tasks and includes the largest collection of NIC models among available libraries. Moreover, our benchmark is easily scalable, enabling systematic evaluation of NIC models in challenging adversarial scenarios.

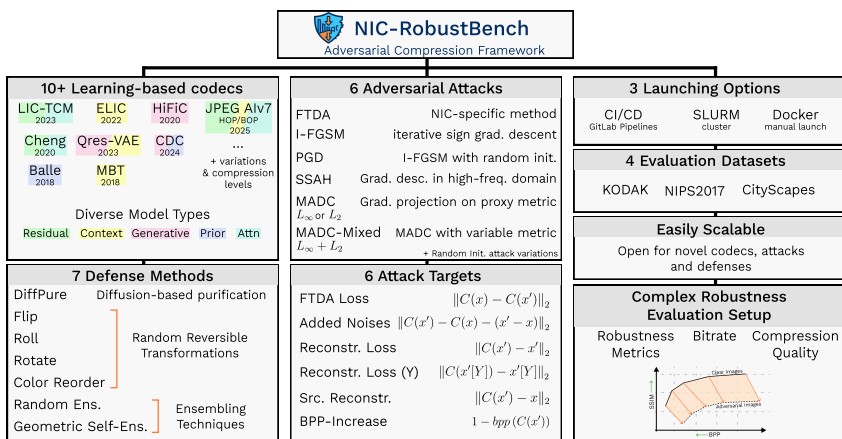

Figure 2: Summary of the contents of our framework.

- **Comprehensive evaluation**: We perform extensive experiments on 10 diverse NIC models (including the latest JPEG AI codec), using 6 different attacks that target either the reconstructed image quality or the compression bitrate. Our evaluation provides valuable insights into the comparative robustness of these models under various attack conditions.

- **Defense strategies**: We investigate several strategies to defend NIC models against adversarial attacks. In particular, we implement and evaluate 7 representative adversarial defense techniques for NIC, and we identify which techniques are most effective at improving the robustness of neural compressors.

## 2 RELATED WORK

**Neural image compression** has seen rapid progress in recent years. Ballé et al. (2016) introduced one of the first models for end-to-end compression with generalized divisive normalization and uniform scalar quantization. Agustsson et al. (2017) proposed a soft-to-hard vector quantization method for compressive autoencoders. Ballé et al. (2018) further modeled image compression as a variational autoencoder problem, introducing hyperpriors to improve entropy modeling. Minnen et al. (2018) extended hierarchical Gaussian scale mixture models with Gaussian mixtures and autoregressive components. Mentzer et al. (2020) utilized GANs to enhance perceptual quality in neural compression. Cheng et al. (2020) developed an efficient entropy model based on discretized Gaussian mixtures and attention modules. Yang et al. (2020) introduced rate and complexity control using slimmable modules. He et al. (2022) proposed the ELIC model, which combines stacked residual blocks with a spatial-channel context entropy model. Zou et al. (2022) enhanced compression with window-based attention, training both CNN and Transformer models. Liu et al. (2023a) combined transformer-CNN mixture blocks with a Swin-Transformer attention module. Duan et al. (2023) adopted a hierarchical VAE with a uniform posterior and a Gaussian-convolved uniform prior. Wang et al. (2023) focused on real-time compression using residual and depth-wise convolution blocks, introducing mask decay and sparsity regularization for model distillation. Yang & Mandt (2024) presented a lossy compression scheme employing contextual latent variables and a diffusion model for reconstruction. The novel JPEG AI standard (Ascenso et al. (2023)) applies learned quantization across the entire image, surpassing traditional block-based methods in efficiency. It offers high and base operation points to balance compression efficiency and computational complexity, and includes adaptive tools depending on codec configuration.

**Adversarial robustness of NIC.** Liu et al. (2023b) proposed a bitrate-oriented adversarial attack on NIC models based on I-FGSM (Kurakin et al. (2017)). Their study explored the impact of this attack across multiple codec architectures, highlighting design components that contribute to enhanced robustness, and identified their proposed factorized attention model as the most stable architecture. Similarly, Chen & Ma (2023b) demonstrated that NIC methods are highly susceptible to adversarial perturbations that reduce the quality of the decoded image. They adopted the I-FGSM and C&W

(Carlini & Wagner (2017)) attacks to increase the difference between the compressed images before and after the attack and introduced the Fast Threshold-constrained Distortion Attack (FTDA) that attacks images after compression. Furthermore, they proposed the $\Delta$PSNR metric for evaluating attack performance, which jointly captures the trade-off between attack effectiveness and perceptual visibility.

# 3 ADVERSARIAL ROBUSTNESS OF NIC

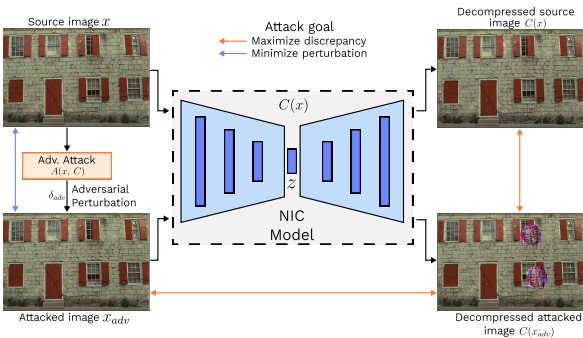

Figure 3: Process of adversarial attack on NIC with possible attack goals.

**NIC models.** Lossy image compression is based on a rate-distortion theory. The goal is to find a trade-off between the size of the compressed image representation and the decrease in the perceptual quality of a reconstructed image:

$$\mathbb{E}_x[\lambda r(\hat{y}) + d(x, \hat{x})], \qquad (1)$$

where $r(\hat{y})$ is a bit representation of a quantized image after arithmetic encoding, and $d(x, \hat{x})$ — perceptual image-similarity metric. Encoder-decoder architecture is one of the possible solutions. For a given image $x \in X = \mathbb{R}^{H \times W \times 3}$, an encoder $E$ transforms it to a latent representation $y = E(x)$. Then, the data is quantized $\hat{y} = Q(y)$, and a decoder $G$ performs the reconstruction of the image $\hat{x} = G(\hat{y})$. We denote $C(x)$ as a complete encoding-decoding process $C(\cdot) = G \circ Q \circ E : X \to X$.

**Adversarial attacks.** The goal of an adversarial attack is to find a perturbation $\delta$ which, added to the original image, makes the adversarial image $x' = x + \delta$ such that its decoded image $C(x')$ differs from the original image as much as possible. Adversarial attack $A : X \to X$ is defined as follows:

$$A(x) = \underset{x':\rho(x',x) \leq \varepsilon}{\arg\max} \ L(x, x', C(x), C(x')), \qquad (2)$$

where $\rho(x', x) = \|\delta\|$, $\varepsilon$ imposes a constraint on the perturbation magnitude, $L : X \times X \to \mathbb{R}$ is a corresponding optimization target. To achieve this goal, we consider 6 loss functions for all employed attacks. They reflect different approaches to measuring the distance between the original and adversarial images and their reconstructed versions. Additionally, we consider an alternative optimization goal. Instead of increasing the distance between the image targets, we reduce the compression ratio of the NIC measured in Bits Per Pixel (BPP):

$$A(x) = \underset{\delta:\|\delta\| \leq \varepsilon}{\arg\max} \text{BPP}(Q(E(x + \delta))). \qquad (3)$$

**Adversarial defenses.** The existing adversarial defenses for NIC models are adversarial training and adversarial purification. In this study, we focus on the second type of defences, as it is more universal. Compared to adversarial training, adversarial purification is simple to apply to ned NIC modes as it does not require model re-training, which may be tricky especially for JPEG AI. Adversarial purification defense consists of two additional steps in the compression procedure: a preprocessing transformation $T : X \to X$ before compression and a postprocessing transformation $T^{-1} : X \to X$ afterward. Formally, an adversarial attack of the defended NIC can be defined as:

$$A(x) = \underset{x':\rho(x',x) \leq \varepsilon}{\arg\max} \ L(x, x', g(x), g(x')), \qquad (4)$$

where $x \in X$ is the original image, $x' \in X$ is the adversarial image, $g(\cdot) = (T^{-1} \circ C \circ T)(\cdot)$ is the adversarially defended NIC, $T(\cdot) : X \to X$ is adversarial defense, $T^{-1}(\cdot) : X \to X$ is the inverse of that defense. This equation can be transformed into an adversarial attack on undefended NIC when $T(X) = X$. Alongside reversible defenses, we also include DiffPure (Nie et al. (2022)), a popular adversarial purification method, which does not have postprocessing step $T^{-1}$. In this case, we use $T^{-1}(X) = X$.

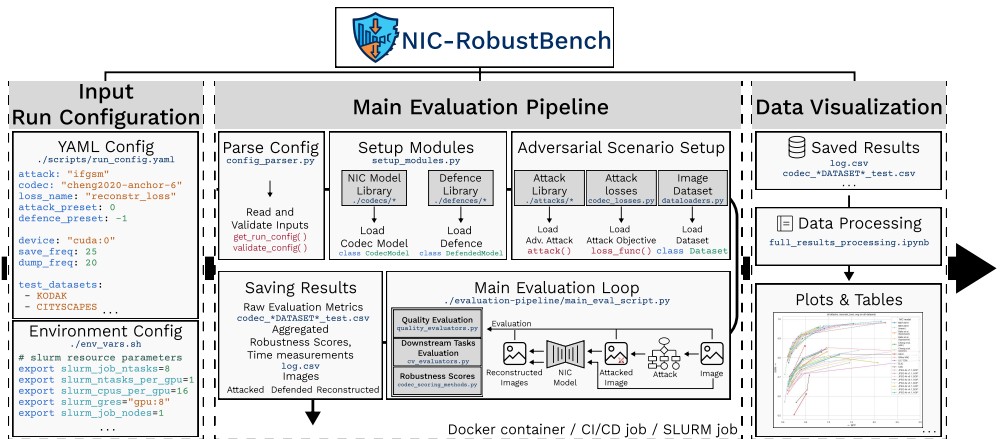

Figure 4: Overview of NIC-RobustBench modular framework structure and NIC evaluation pipeline.

## 4 OVERVIEW OF NIC-ROBUSTBENCH

NIC-RobustBench is a research-oriented framework designed for systematic and comprehensive evaluation of NIC models using standardized pipelines. It provides a flexible and easily scalable test ground for further research at the intersection of adversarial robustness and learned image compression fields. Core use cases of our library include:

- **General codec compression efficiency evaluation** in terms of image quality, compression ratio, model speed, number of parameters, and downstream CV tasks performance.

- **Attack simulation and NIC Robustness benchmarking** in a broad spectrum of adversarial scenarios with attacks using various optimization objectives that impact the compression-decompression pipeline differently.

- **Development and testing of defensive mechanisms** designed to alleviate the impact of attacks on codec performance.

Figure 2 (b) summarizes the contents of our framework. It includes the list of implemented NIC types, adversarial attacks, optimization targets for them and adversarial defenses to counter the attacks. Our framework builds upon and extends prior research on the robustness of neural image compression methods, implementing a record number of NIC models, NIC-specific attacks and defenses compared to existing open-source libraries (Table 1).

Figure 4 demonstrates the modular design of the NIC-RobustBench framework and its step-by-step evaluation pipeline. Modular architecture of our library simplifies code structure and allows for effortless scaling by implementing NIC models, datasets, adversarial attacks and defenses as standardazied classes and functions with unified methods and input arguments.

Table 1: List of popular NIC libraries and NIC robustness evaluation methods

| Paper | # NIC types\variants | # attacks | # defenses |
|---|---|---|---|
| NIC robustness studies | | | |
| Liu et al. (2023b) | 2\42 | 2 | 1 |
| Chen & Ma (2023b) | 8\42 | 3 | 3 |
| Libraries | | | |
| CompressAI (Bégaint et al. (2020)) | 3\36 | 0 | 0 |
| TFC (Ballé et al. (2024)) | 5\- | 0 | 0 |
| NeuralCompression (Muckley et al. (2021)) | 5\- | 0 | 0 |
| OpenDIC (Gao et al. (2024)) | 7\- | 0 | 0 |
| NIC-RobustBench (Proposed) | **10\47** | **6 ($\times$ 6 objectives)** | **7** |

Main robustness evaluation pipeline consists of several stages. First, it parses the input configuration, loads a selected NIC model, defensive algorithm and dataset, and sets up the adversarial scenario. For each dataset image, a corresponding adversarial example is generated, and both images are passed through the compression-decompression process. Next, images before and after NIC pass through the evaluators that track image quality, compression efficiency, time and downstream

performance of different computer vision models. Finally, raw metric values are aggregated into evaluation scores, and all results are saved with the corresponding run metadata.

All stages of the pipeline are easily configurable and customizable, as core run parameters can be specified via single YAML configuration file. By leveraging Docker-based containerization and simple YAML-based setup configuration, NIC-RobustBench aims for experiment reproducibility. Additionally, NIC-RobustBench includes tools for results visualisation, which allows researchers to quickly obtain a visual summary of the experiments.

## 5 NIC ROBUSTNESS BENCHMARK

**NIC models.** Our framework includes most of the available open-source NIC models, including LIC-TCM (Liu et al. (2023a)), ELIC (He et al. (2022)), HiFIC (Mentzer et al. (2020)), JPEG AI (Ascenso et al. (2023)), Cheng et al. (Cheng et al. (2020)), EVC (Wang et al. (2023)), Qres-VAE (Duan et al. (2023)), Balle et al. (Ballé et al. (2018)), MBD (Minnen et al. (2018)) and CDC (Yang & Mandt (2024)). More details on this models can be found in Section 2.

**Adversarial attacks.** We focus exclusively on white-box attacks, since black-box methods require a large number of queries to the target model to reach performance comparable to white-box attacks, making them computationally inefficient and impractical for benchmarking. As summarized in Fig. 2 (right), We choose six different white-box attacks of various types. **MADC** (Wang & Simoncelli (2008)) was one of the first methods that uses gradient projection onto a proxy Full-Reference metric to preserve image quality. We employ 3 variations of MADC attack with different proxy metrics: $L_2$, $L_\infty$ and their combination, which we name MADC-Mixed. **I-FGSM** (Kurakin et al. (2017)) is a well-known iterative modification of FGSM attack with simple sign gradient descent. **PGD** (Madry et al. (2018)) is similar to I-FGSM but uses random initialization. FTDA (Chen & Ma (2023b)) attack is specifically designed to target neural image compression models. Additionally, we adopt frequency-based **SSAH** (Luo et al. (2022)) attack from the classification field for learned image compression. It decomposes an image into low- and high-frequency domains and insert perturbation in the latter to reduce attack visibility.

**Adversarial defenses.** We selected several reversible adversarial defenses to evaluate their efficiency against adversarial attacks on NIC. We employed mainly reversible transformations to reduce their effect on image degradation. **Flip** takes an image and reflects it horizontally or vertically. The reversed version flips the output image again to restore the original image orientation. **Random Roll** selects either height or width randomly and samples the size of the roll, then rolls the image by a random number of pixels. The reverse step restores the original alignment. **Random Rotate** defense samples the angle randomly and rotates the image on the selected angle. To make restoration of the original image possible, the method also performs a center pad of the original image to ensure image borders do not cut all its content. **Random Color Reorder** defense chooses perturbation of the color channels of the image tensor and swaps them, restoring the original order of the NIC output. **Random Ensemble** combines Roll, Rotate, and Color reorder properties. It samples 10 actions from Roll, Rotate, and Color reorder with 4, 4, and 1 weights, respectively. **Geometric Self-Ensemble** (Chen & Ma (2023a)) generates 8 defended image candidates with flipping and rotation. It chooses one of them as the output that resembles the least distorted after the preprocess-NIC-postprocess pipeline. The distortion is measured by the mean squared error between the original image and the image processed by the defended NIC. **DiffPure** (Nie et al. (2022)) was developed to counter adversarial attacks on image classifiers and showed state-of-the-art performance for defending computer vision models. It performs purification based on a diffusion model as preprocessing and does nothing in the postprocessing step.

**Datasets.** To evaluate methods, we chose three well-known datasets. The KODAK Photo CD (Kodak (1991)) dataset consists of 24 uncompressed images, each with a resolution of $768 \times 512$. We sample 50 $512 \times 256$ images from CITYSCAPES (Cordts et al. (2016)), which is a domain-specific dataset for image segmentation tasks. These datasets are commonly used in the field of image compression. Additionaly, NIPS 2017: Adversarial Learning Development Set (Kurakin et al. (2018)) contains 1000 $299 \times 299$ images and is designed for evaluating adversarial attacks against image classifiers.

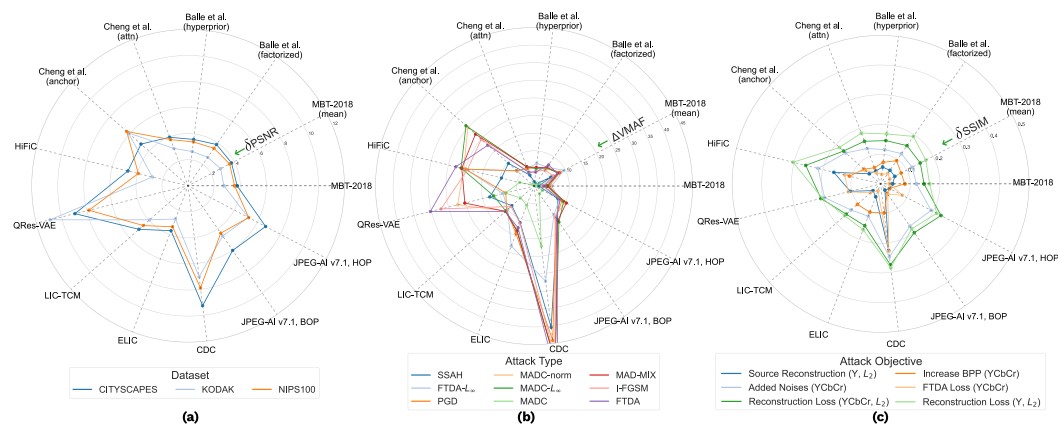

Figure 5: Robustness evaluation of tested NIC models measured across (a) different datasets, (b) different attacks, and (c) different attack objectives.

**Evaluation metrics.** We employ four different Full-Reference image quality metrics to numerically assess the effects of adversarial attacks on images before and after the reconstruction: PSNR, MSE, MS-SSIM (Wang et al. (2003)) and VMAF (Li et al. (2018)). PSNR, MSE, and MS-SSIM are traditional image-similarity measures. MS-SSIM Wang et al. (2003) provides a scale-independent quality estimate, and VMAF implements a learning-based approach that aligns well with human perception (Antsiferova et al. (2022)). VMAF was designed to estimate the quality of distorted videos, but it can also be applied to images, interpreting them as single-frame videos.

Following the methodology of Chen & Ma (2023b), we measure $\Delta_{score}$ , which quantifies how adversarial distortion changes after image reconstruction by an NIC model:

$$\Delta_{score} = FR(x_i, x_i') - FR(C(x_i), C(x_i')), \tag{5}$$

where $x$ is an original image, $x'$ is a corresponding adversarial example, $FR(x, y)$ is one of the aforementioned Image Quality Assessment models, and $C(x)$ is an evaluated NIC model (entire encoding-decoding procedure). If higher values of considered $FR$ model correspond to better visual quality (e.g., for PSNR, MS-SSIM, and VMAF), then positive $\Delta_{score}$ indicate that the input perturbation was amplified by the model, whereas negative values indicate that it was suppressed.

Additionally, we introduce $\delta_{score}$ — a slightly different measure that captures the difference in reconstruction quality between the clear image and corresponding adversarial example:

$$\delta_{score} = FR(x, C(x)) - FR(x', C(x')), \tag{6}$$

Higher values of $\delta_{score}$ indicate that adversarial example introduced significant distortions to the decompressed image after passing through the NIC. Conversely, near-zero values suggest that quality loss due to compression was roughly equal between the clear image and its adversarial counterpart.

## 6 EVALUATION RESULTS

### 6.1 NIC PERFORMANCE EVALUATION

The main results of NIC performance evaluation are presented in Fig. 5 and 6. Fig. 5 illustrates the impact of adversarial attacks on the reconstruction quality of neural compression models across different data slices. Fig. 6 compares the performance of neural codecs on clean and adversarially perturbed data under different target objectives used in the attacks. Based on these results, the following insights can be drawn.

**Generative codecs are among the most vulnerable to attacks.** Codecs that employ generative priors in their design, namely CDC, HiFiC and QRes-VAE demonstrate high $\Delta$ and $\delta$ scores (Fig. 5), suggesting stronger adversarial effects. HiFiC uses GAN in its architecture to train adversarially against discriminator, CDC employs a diffusion model, and QRes-VAE uses Variational AutoEncoder. However, we also note that this effect might be attributed to the larger size of these models compared to other codecs, as described below.

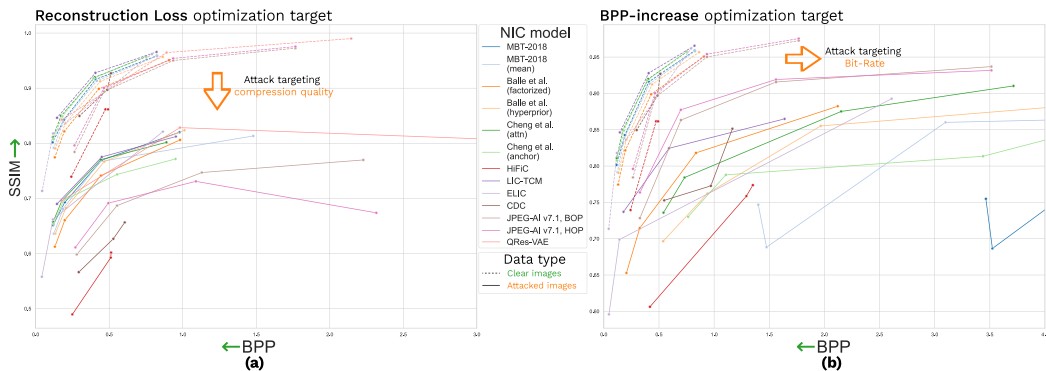

Figure 6: RD-curves representing NIC performance on clear (dashed lines) and attacked (solid) data. Attacks target image quality on the left subfigure, and bitrate on the right.

**Across NIC model families, larger models tend to be less secure**. Throughout our study, we found a significant (Spearman Corr. $0.724$, $p < 10^{-8}$) positive correlation between model size and average efficiency of the attacks, as presented in Fig. 7. While the scale of the effect varies from model to model, larger parameter count appears to create more pathways within the model that attacks could potentially exploit. Fig. 5 also confirms that smaller Balle et al. models and MBT-2018 are among the most resilient, while larger HiFiC and CDC are among the most vulnerable.

**Within a NIC model family, those with higher compression rates are the most robust**. These models (with lower BPPs, see variants 0 and 1 in Fig. 7) typically have fewer parameters and are thus less vulnerable. Moreover, smaller models targeting stronger compression often introduce image blur by prioritizing lower frequencies over higher ones, which can "erase" adversarial perturbations more effectively during compression. Therefore, this effect may be attributed not only to the reduced number of parameters but also to the frequency characteristics of the compression process.

**Robustness scores are mostly consistent across images with different resolutions, aspect ratios, and content**. The evaluation of NIC models on three distinct datasets, presented in Fig. 5 (a), revealed a nearly identical robustness profile, with no substantial differences observed across datasets.

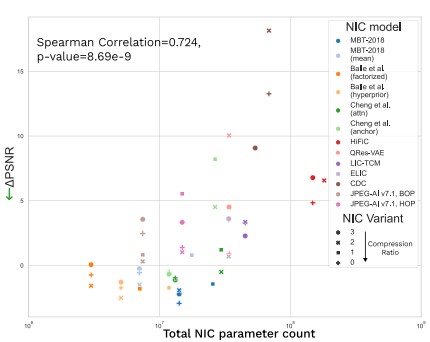

Figure 7: Relationship between NIC size, compression ratio, and robustness.

**Best-performing NICs in non-adversarial scenario vary across different bitrates.** Alongside NIC robustness comparison, our framework can also be used to directly compare NIC models on non-attacked images, like other NIC libraries. Dashed rate-distortion (RD) curves in Figure 6 represent the performance of different NICs on source images. In the lowest bitrate range ($< 0.2$ BPP), context-based ELIC model leads, as it reaches the lowest bitrate among tested models. In the most competetive lower-to-mid bitrate range ($0.2 - 0.8$ BPP), NIC models that incorporate attention mechanisms, namely LIC-TCM and Cheng2020-attn, show the best efficiency in terms of SSIM. LIC-TCM combines transformer blocks with CNN backbone, and Cheng2020-attn enhances a residual network with Self-Attention. For the highest bitrate scenarios, VAE-based QRes-VAE model shows the best quality.

## 6.2 ATTACK PERFORMANCE COMPARISON

**Distinct attack objectives have vastly different impacts on the target NIC model.** Fig. 6 shows the effects of attacks targeting two "orthogonal" measures of codec performance: rate (right subfigure) and distortion (left). In both cases, all NIC models suffer significant performance drop; however, the direction of the impact differs. Reconstruction attack objective severely impairs the quality of reconstructed images compared to their uncompressed variants, shifting RD-curves downwards. BPP-increase objective, on the other hand, targets compressed file size, and shifts RD-curves to

the right, towards higher BPP. At the same time, models that are robust against attacks based on one objective function may become vulnerable when a different objective is used. For example, the MBT-2018 family demonstrated strong robustness under the Reconstruction attack objective but proved to be the most vulnerable under the Increase-BPP target.

**Attacks directly targeting image reconstruction have the strongest impact on NIC performance in terms of image quality.** As illustrated in Fig. 5 (c), attack objectives that maximize the distance between attacked images before and after compression ("Reconstruction Loss" in Fig. 5, i.e. $||C(x') - x'||_2$) result in the highest loss of SSIM. Other targets that include clear image $x$ or its decompressed variant $C(x)$ to the objective show weaker impact on quality. Moreover, the most effective variant is reconstruction loss applied only to the luminance channel, which enabled stronger disruption of the reconstructed image structure.

**Attack efficiency varies greatly between different codecs.** As Fig. 5 (b) demonstrates, FTDA and MADC-$L_\infty$ are among the strongest in our study, showing highest $\Delta$ scores on most codecs. However, two exceptions are notable: on Cheng2020 (anchor) model, FTDA efficiency drops significantly, while MADC-based attacks and I-FGSM perform well. On QRes-VAE, conversely, FTDA significantly outperforms MADC-$L_\infty$. This effect might be due an important difference in attack designs: MADC-$L_\infty$ attack quantizes the model gradient with a $sign$ operation, while FTDA uses unmodified gradient in the optimizer. This gradient quantization step might speed up the attack convergence for some models (e.g., Cheng2020 (anchor)), while significantly impairing the accuracy of the optimization steps for others (e.g., QRes-VAE). This emphasizes that NIC models should be tested against various attack designs to ensure better robustness.

### 6.3 Defense evaluation

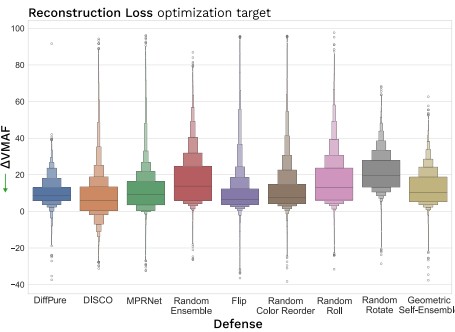

Figure 8: Performance of different defensive algorithms.

**Diffpure is among the strongest adversarial defenses.** For DiffPure $\Delta$VMAF is more concentrated compared to other defenses and lacks outliers at the top, hinting that it is more stabilizing and predictable for NIC models. DiffPure helps most likely due to the fact that it actually removes off-manifold, high-frequency adversarial noise and pulls inputs back toward the data distribution.

**Geometric transformations without intrinsic interpolation work better**. A simple random flip works well because it's a lossless transform that matches the model's learned invariance from training, so perturbations don't transfer while semantics stay intact. In contrast, rotate and roll introduce interpolation/edge artifacts and a distribution shift the model isn't equivariant to, hurting clean accuracy and capping robust gains.

### 6.4 Additional results

We provide more detailed results in the Appendix, including attack transferability evaluation between different NIC models (Sec. A.1), computational performance evaluations (Sec. A.2), statistical tests (Sec. A.3), attacks on downstream computer vision tasks (Sec. A.7) and other experiments.

## 7 Conclusion

This paper proposes NIC-RobustBench, a novel open-source library that can be used by researchers for evaluating NIC robustness. To our knowledge, NIC-RobustBench is the first framework that implements a comprehensive collection of adversarial attacks and defenses for a learned image compression task. It includes the largest number of NIC models among the open-source libraries known to the authors. NIC-RobustBench provides an efficient, simple and scalable framework to evaluate the performance, robustness and compression quality of NIC models. Utilizing NIC-RobustBench,

we reveal systematic degradation of compression quality and efficiency in most modern learning-based codecs under adversarial attacks, emphasizing the importance of robustness validation in image-compression task.

## 8 REPRODUCIBILITY STATEMENT

We attach the full code for the experiments, pre-trained checkpoints, and a Docker image in the supplementary and on a public repository upon acceptance.

All stages of the pipeline are easily configurable and customizable, as core run parameters can be specified via single YAML configuration file. By leveraging Docker-based containerization and simple YAML-based setup configuration, NIC-RobustBench aims for experiment reproducibility. Additionally, NIC-RobustBench includes tools for results visualisation, which allows researchers to quickly obtain a visual summary of the experiments.

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

# A APPENDIX

## A.1 TRANSFERABILITY OF ADVERSARIAL EXAMPLES

This section presents the transferability of adversarial examples generated for one NIC to others. We compare different bitrates of chosen codecs, so they are included several times. We ran experiments on Reconstruction Loss for two presets and averaged the results. The results from figure 9 show high transferability between different bitrates of specific versions of each NIC model, especially from lower bitrates to higher ones. There is also transferability from a more stable NIC model to a less stable one.

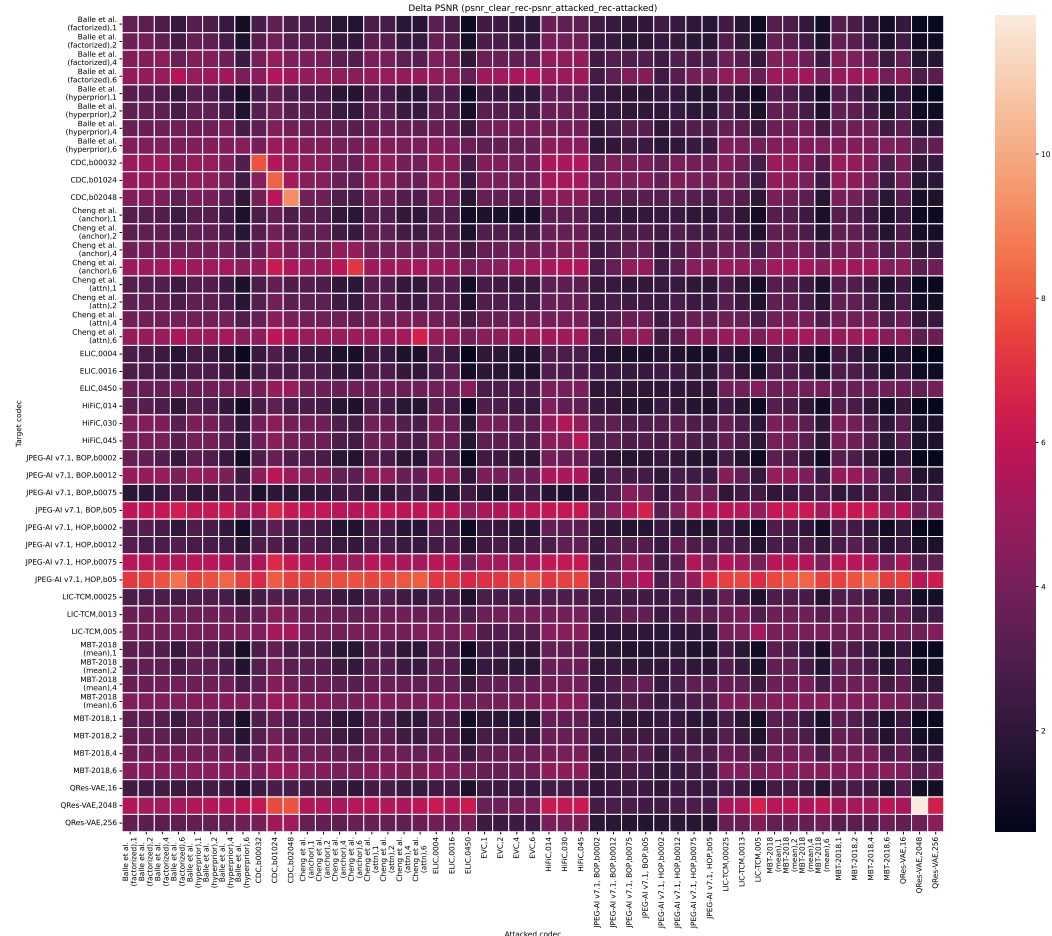

Figure 9: Transferability of adversarial attacks constructed for codecs listed in columns to other codecs (listed in rows). The metric for attack success is $Delta$ PSNR.

## A.2 SPEED COMPARISON

Figure 10 compares complexity of different NIC models. JPEG AI prioritize accuracy/quality, but among the slowest and most memory-hungry in practice.

- Peak memory varies by $\sim \times 10$. Several pipelines (e.g., JPEG-AI BOP/HOP, CDC) peak around 3–4 GB, while classical baselines (factorized / hyperprior) stay well below 1 GB.

- HiFiC carries the largest parameter budget ($\sim 10^7$) yet is among the fastest; conversely, several mid-sized models run slowly. Compute is dominated by architecture and operator choice (context models, attention/masked convs, entropy loops), not just parameter size.

- Memory and latency are correlated, but not monotonically. Attention/context-heavy designs generally incur both high memory and time; some GAN/hyperprior models decouple the two (large params, low latency).

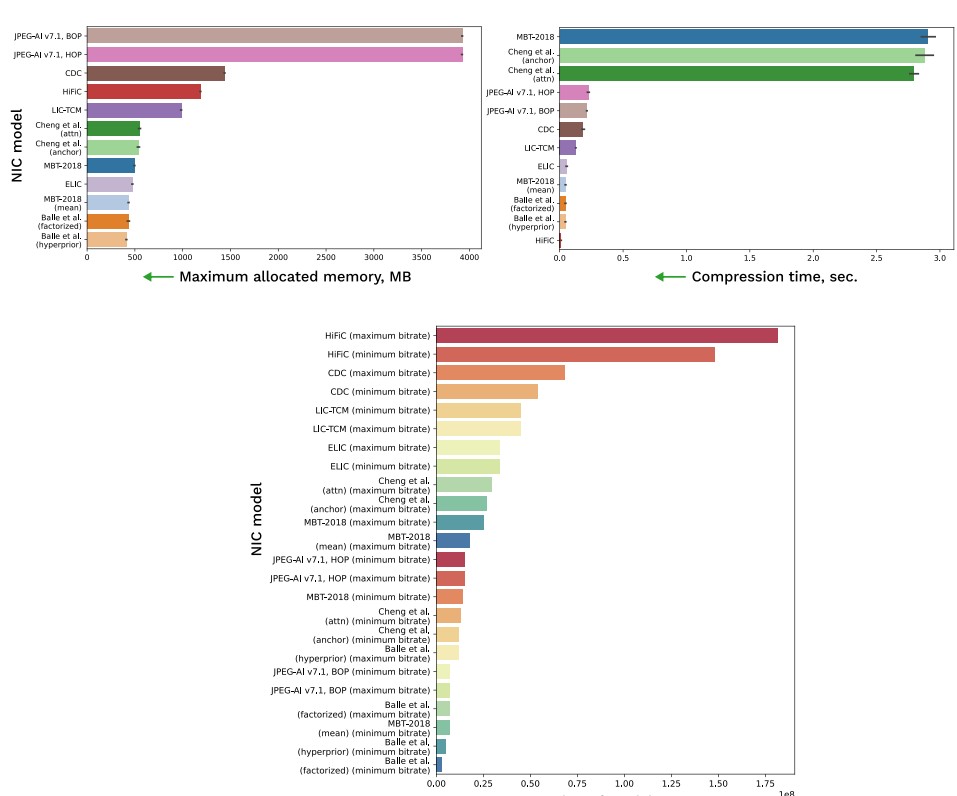

Figure 10: Computational complexity of different NIC models measured in maximum allocated memory during the compression-decompression process (a), compression time (b), and number of model parameters (c).

## A.3 STATISTICAL TESTS

We applied the one-sided Wilcoxon Signed Rank Test to assess the statistical significance of codec comparisons, as it is non-parametric and suited for paired samples without assuming normality (which restricts potential distributions of evaluation scores) — ideal for adversarial robustness analysis. This test evaluates whether one NIC consistently outperforms another in terms of $\Delta$ scores and other relevant metrics. Results for $\Delta$SSIM and $\delta$SSIM scores are provided in Figures 12 and 11 respectively. To increase readability, we only use 2 compression ratios for each NIC model (out of 4), the highest and lowest.

To ensure reliability, we applied the Bonferroni correction, which controls the family-wise error rate across hundreds of comparisons (13 codecs$\times$ 2 compression ratio per codec, squared). This conservative adjustment minimizes false positives, reinforcing the significance of the results.

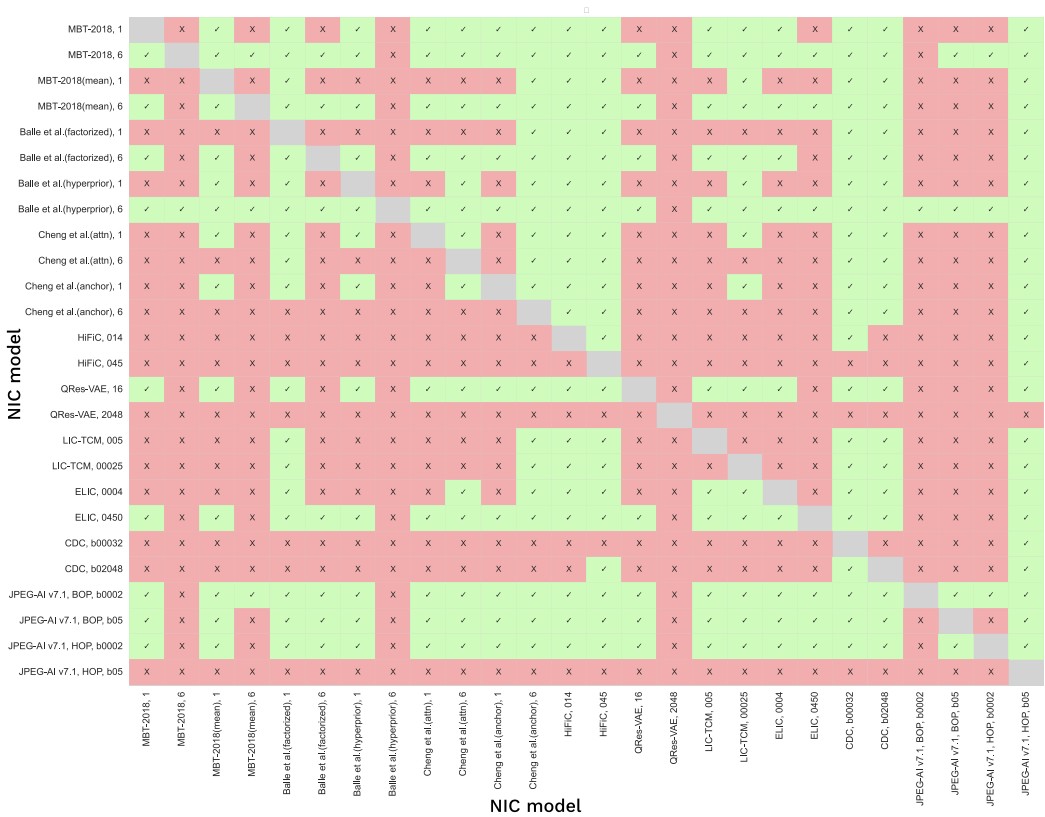

Figure 11: Results of Wilcoxon signed-rank tests between different NICs based on $\delta$**SSIM** scores. ✓ symbols represents cases when NIC model denoted in the row statistically outperforms the NIC defined in the column with a $p$-value$< 0.05$. Bonferroni correction is used to account for large number of pair-wise comparisons. In this experiment, we employ all attacks with "Reconstruction Loss" objective.

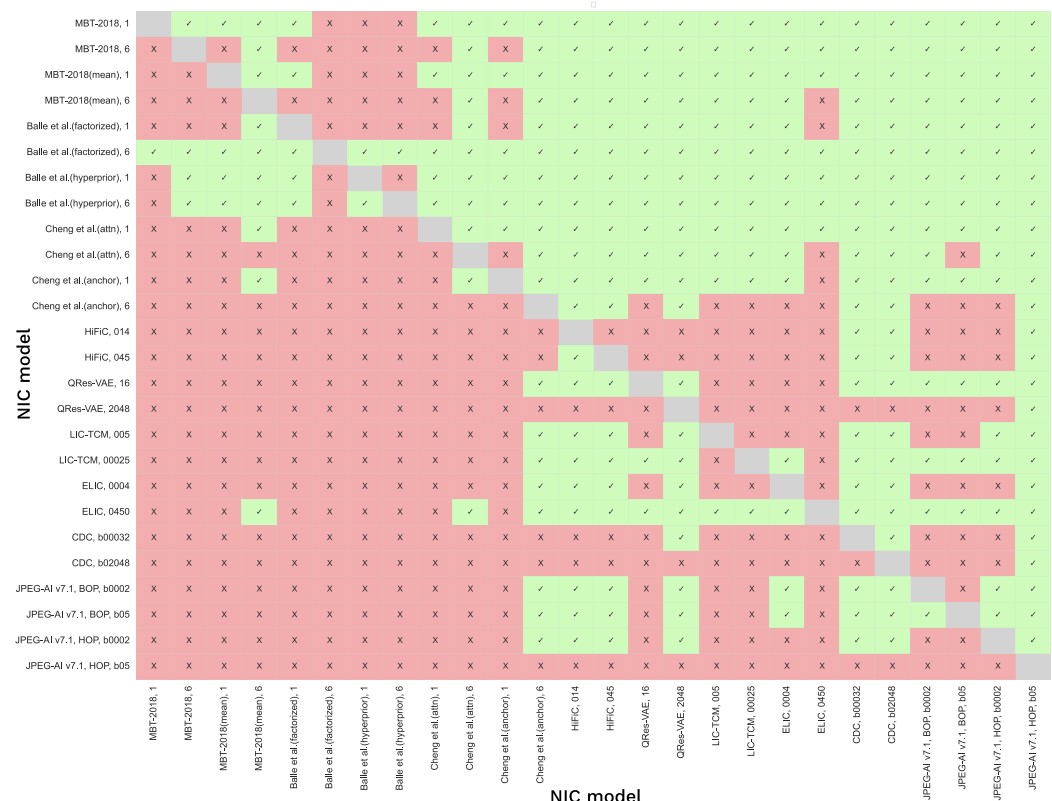

Figure 12: Results of Wilcoxon signed-rank tests between different NICs based on Δ**SSIM** scores. ✓ symbols represents cases when NIC model denoted in the row statistically outperforms the NIC defined in the column with a $p$-value$< 0.05$. Bonferroni correction is used to account for large number of pair-wise comparisons. Here, we employ all attacks with "Reconstruction Loss" objective.

## A.4 PARAMETER SENSITIVITY

Fig. 13 shows effectiveness of attacks depending on the learning rate. The effectiveness of attack methods can vary greatly depending on the specific attack parameters chosen for each codec. To ensure that the attacks function correctly, their parameters are selected based on a grid of potential values for each codec individually. Afterwards, presets (sets of attack parameters) are created that are suitable for a wide range of codecs. For example, for attacks based on MADC with a single learning rate parameter $\{lr = 0.01\}$ proved to be sufficient, as it performed well across all codecs.

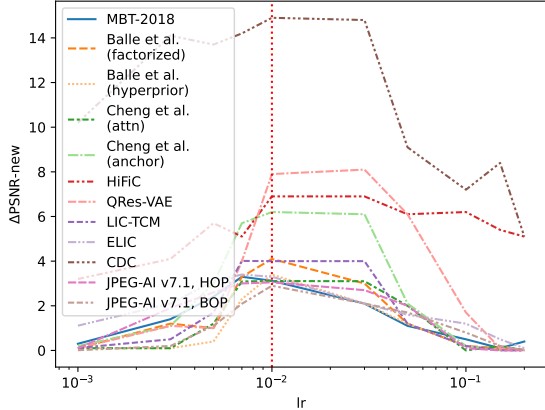

Figure 13: Comparison of the effectiveness of MADC family attacks depending on the learning rate

## A.5 RELATIONSHIP BETWEEN $\delta$ AND $\Delta$ ROBUSTNESS SCORES

Fig. 14 illustrates the relationship between different methods for calculating quality degradation after an attack. While $\Delta$ and $\delta$ are calculated between various pairs of clean, adversarial, decoded, and adversarial decoded images, their values generally correlate, and the lists of best- and worst-performing codecs are mostly similar between the two. The highest Spearman correlation between $\delta$ and $\Delta$ is 0.783 using VMAF as a quality metric. These results indicate that the proposed $\delta$ metric is a reasonable approach to measuring adversarial stability. The lower correlation for other FR metrics suggests that $\delta$ and $\Delta$ capture different aspects of the adversarial robustness of NIC models.

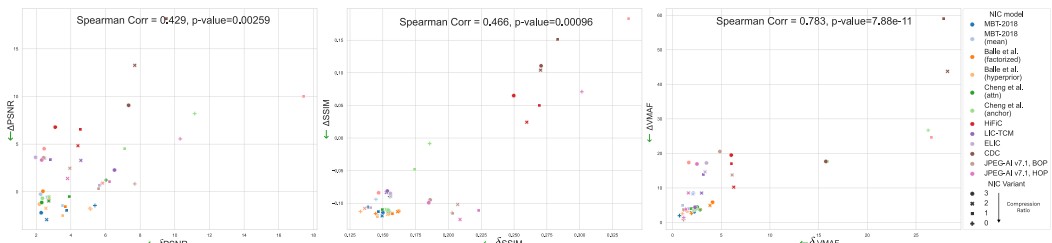

Figure 14: Relationships between $\Delta$ and $\delta$ scores for different FR quality assessment metrics.

## A.6 ADVERSARIAL DEFENSES PARAMETERS

Table 2 presents a description of the adversarial defenses integrated into the framework, including their parameters for the forward and inverse transformations.

| Defense method | Type | Parameters | Preprocess $(Y = T(X))$ | Postprocess $(X = T^{-1}(Y))$ |
|---|---|---|---|---|
| Flip | Spat. transf. | — | $\text{flip}(X, [2, 3])$ | $\text{flip}(Y, [2, 3])$ |
| Random roll | Spat. transf. | $\dim \in \{2, 3\}$, $\text{size} \in [0, \text{len}(X[\dim]) - 1]$ | $\text{roll}(X, \text{size}, \dim)$ | $\text{roll}(Y, -\text{size}, \dim)$ |
| Random rotate | Spat. transf. | $\theta \in [0, 359]$ | $\text{pad}(\text{rotate}(X, \theta))$ | $\text{crop}(\text{rotate}(Y, -\theta))$ |
| Random color reorder | Color transf. | $\sigma : \{0, 1, 2\} \to \{0, 1, 2\}$ | $X[:, \sigma([0, 1, 2])]$ | $Y[:, \sigma^{-1}([0, 1, 2])]$ |
| Random ens. | Ensemble | — | Varies | Varies |
| Geometric self-ens. (Chen & Ma (2023a)) | Ensemble | — | Varies | Varies |
| DiffPure (Nie et al. (2022)) | Purification | — | $\text{diffpure}(X)$ | $Y$ |

Table 2: List of adversarial defenses used in our paper.

## A.7 ATTACKS ON DOWNSTREAM COMPUTER VISION TASKS

Our paper aims to evaluate the impact on the various computer vision (CV) tasks, notably classification, detection and depth estimation. The evaluation can be described by the following idea: we feed the attacked image to the CV model, validate relevant metrics and compare them with the original scores. For the given image $x \in X$ and CV model $F : X \to S$, where $S$ is the output of the CV model (logits in case of image classification, bounding box coordinates in case of detection and $\mathbb{R}^{H \times W}$ in case of depth estimation), we can describe the evaluation by the following equation:

$$s = F(\arg\max_{x':\rho(x',x)\leq\varepsilon} L(x, x', C(x), C(x'))) \tag{7}$$

where $x \in X$ is the original image, $s \in S$ is the output of the CV model.

ImageNet (Deng et al. (2009)) for classification, MS COCO (Lin et al. (2014)) for detection, KITTI Depth (Uhrig et al. (2017)) for depth prediction are used for testing downstream tasks due to the availability. The evaluation metrics used are common for the relevant tasks: accuracy for classification, precision, recall and f1 for classification and detection, IoU for detection, MAE for depth estimation.

We evaluate mentioned metrics and demonstrate the difference between clean and attacked recinstructed images averaged by attacks in Table A.7. We have taken ResNet50 (He et al. (2016)) for classification, YOLO11 (Jocher & Qiu (2024)) for detection and Depth Anything V2 (Yang et al. (2024)) for depth estimation. The results demonstrate that classification and depth estimation are rather robust to adversarial attacks on NIC models. However, the task of detection is significantly impacted by adversarial attacks.

Adversarial examples from ELIC model are the most efficient in attacking the classification model, while examples from QRes-VAE are efficient in attacking detection and depth estimation models.

| NIC | Classification | | | | Detection | | | | Depth Estimation |
|---|---|---|---|---|---|---|---|---|---|
| | $\Delta$Accuracy | $\Delta$Precision | $\Delta$Recall | $\Delta$F1 | $\Delta$Precision | $\Delta$Recall | $\Delta$F1 | $\Delta$IoU | $\Delta$MAE |
| Balle et al.(factorized), 1 | -0.020 | -0.024 | -0.019 | -0.023 | 0.018 | 0.019 | 0.019 | -0.004 | -0.001 |
| Balle et al.(factorized), 2 | 0.010 | 0.005 | 0.004 | 0.005 | 0.024 | 0.043 | 0.033 | -0.006 | -0.001 |
| Balle et al.(factorized), 4 | -0.030 | -0.036 | -0.037 | -0.036 | 0.001 | 0.004 | 0.002 | 0.007 | 0.002 |
| Balle et al.(factorized), 6 | -0.005 | -0.006 | -0.004 | -0.005 | 0.023 | 0.004 | 0.015 | -0.013 | 0.001 |
| Balle et al.(hyperprior), 1 | -0.005 | -0.007 | -0.002 | -0.005 | -0.034 | -0.034 | -0.031 | -0.012 | 0.000 |
| Balle et al.(hyperprior), 2 | -0.003 | -0.008 | -0.004 | -0.007 | -0.049 | -0.025 | -0.036 | -0.000 | -0.002 |
| Balle et al.(hyperprior), 6 | -0.005 | -0.004 | -0.004 | -0.004 | -0.006 | -0.006 | -0.007 | -0.004 | -0.001 |
| CDC, b00032 | 0.003 | 0.008 | 0.010 | 0.009 | 0.013 | -0.024 | -0.006 | -0.004 | 0.005 |
| CDC, b01024 | -0.017 | -0.026 | -0.023 | -0.025 | -0.014 | -0.019 | -0.016 | -0.037 | 0.003 |
| CDC, b02048 | -0.069 | -0.078 | -0.076 | -0.077 | -0.062 | -0.076 | -0.070 | -0.020 | 0.004 |
| Cheng et al.(anchor), 1 | -0.008 | -0.010 | -0.006 | -0.008 | -0.045 | -0.013 | -0.030 | 0.003 | 0.002 |
| Cheng et al.(anchor), 2 | 0.010 | 0.008 | 0.007 | 0.008 | -0.045 | -0.037 | -0.038 | 0.005 | 0.002 |
| Cheng et al.(anchor), 4 | -0.030 | -0.030 | -0.031 | -0.031 | -0.042 | -0.037 | -0.038 | -0.047 | 0.006 |
| Cheng et al.(anchor), 6 | -0.057 | -0.054 | -0.052 | -0.053 | -0.085 | -0.073 | -0.081 | -0.073 | 0.017 |
| Cheng et al.(attn), 1 | -0.013 | -0.009 | -0.009 | -0.009 | -0.034 | -0.009 | -0.021 | 0.007 | -0.001 |
| Cheng et al.(attn), 2 | -0.020 | -0.024 | -0.019 | -0.022 | -0.002 | -0.021 | -0.016 | -0.004 | 0.000 |
| Cheng et al.(attn), 4 | 0.000 | -0.000 | -0.002 | -0.001 | -0.014 | -0.002 | -0.007 | -0.004 | 0.004 |
| Cheng et al.(attn), 6 | -0.026 | -0.032 | -0.034 | -0.033 | 0.010 | 0.006 | 0.008 | -0.009 | 0.002 |
| ELIC, 0004 | **0.030** | 0.024 | 0.028 | 0.025 | -0.022 | -0.013 | -0.020 | -0.040 | 0.002 |
| ELIC, 0016 | -0.040 | -0.023 | -0.026 | -0.024 | 0.046 | 0.048 | 0.043 | -0.016 | 0.005 |
| ELIC, 0450 | 0.020 | 0.027 | **0.029** | **0.028** | 0.013 | 0.012 | 0.014 | -0.001 | 0.002 |
| HiFiC, 014 | -0.061 | -0.060 | -0.054 | -0.058 | -0.073 | -0.064 | -0.065 | -0.161 | -0.002 |
| HiFiC, 030 | -0.012 | -0.012 | -0.012 | -0.012 | -0.025 | -0.020 | -0.023 | -0.022 | 0.001 |
| HiFiC, 045 | -0.020 | -0.023 | -0.024 | -0.023 | 0.009 | -0.006 | -0.000 | -0.003 | 0.001 |
| LIC-TCM, 00025 | -0.044 | -0.044 | -0.044 | -0.044 | -0.026 | -0.011 | -0.016 | -0.009 | 0.002 |
| LIC-TCM, 0013 | -0.013 | -0.018 | -0.019 | -0.018 | **0.049** | 0.024 | 0.036 | -0.015 | 0.005 |
| LIC-TCM, 005 | 0.021 | **0.027** | 0.023 | 0.026 | -0.067 | -0.075 | -0.071 | -0.010 | -0.003 |
| MBT-2018, 1 | 0.007 | 0.001 | 0.008 | 0.003 | 0.005 | -0.006 | 0.000 | -0.004 | 0.001 |
| MBT-2018, 2 | -0.030 | -0.031 | -0.029 | -0.030 | 0.012 | -0.006 | 0.005 | **0.008** | -0.002 |
| MBT-2018, 4 | 0.000 | 0.001 | 0.003 | 0.001 | 0.018 | 0.021 | 0.021 | -0.013 | -0.004 |
| MBT-2018, 6 | 0.003 | 0.008 | 0.008 | 0.008 | 0.007 | 0.008 | 0.008 | -0.004 | -0.000 |
| MBT-2018(mean), 1 | -0.035 | -0.034 | -0.036 | -0.035 | -0.017 | -0.019 | -0.018 | -0.002 | 0.000 |
| MBT-2018(mean), 2 | 0.006 | 0.012 | 0.010 | 0.011 | 0.032 | 0.003 | 0.016 | -0.007 | 0.001 |
| MBT-2018(mean), 4 | 0.006 | 0.008 | 0.007 | 0.008 | -0.038 | -0.001 | -0.019 | 0.003 | -0.002 |
| MBT-2018(mean), 6 | 0.006 | 0.011 | 0.013 | 0.012 | -0.018 | -0.000 | -0.008 | -0.002 | -0.001 |
| QRes-VAE, 16 | -0.020 | -0.026 | -0.020 | -0.024 | -0.022 | **0.094** | 0.044 | -0.002 | 0.007 |
| QRes-VAE, 2048 | -0.187 | -0.152 | -0.149 | -0.151 | -0.157 | -0.188 | -0.173 | -0.195 | **0.035** |
| QRes-VAE, 256 | 0.010 | 0.015 | 0.019 | 0.016 | 0.041 | 0.047 | **0.045** | -0.007 | 0.006 |

Table 3: Comparison of results under adversarial scenarios after compression for downstream tasks (image classification, object detection and depth estimation).

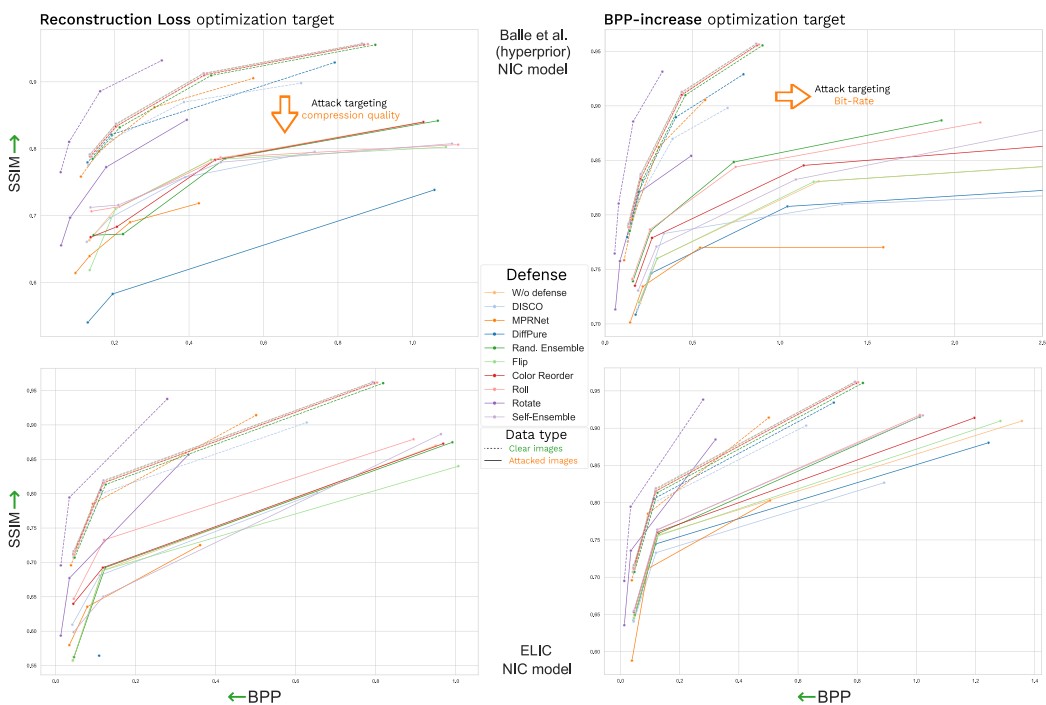

Figure 15: RD-curves representing NIC performance on clear (dashed lines) and attacked (solid) data with different defenses as image preprocessing. First row represents results for Balle et al. (hyperprior) NIC model, and the second row — for ELIC model. Attacks target image quality on the left subfigure, and bitrate on the right.

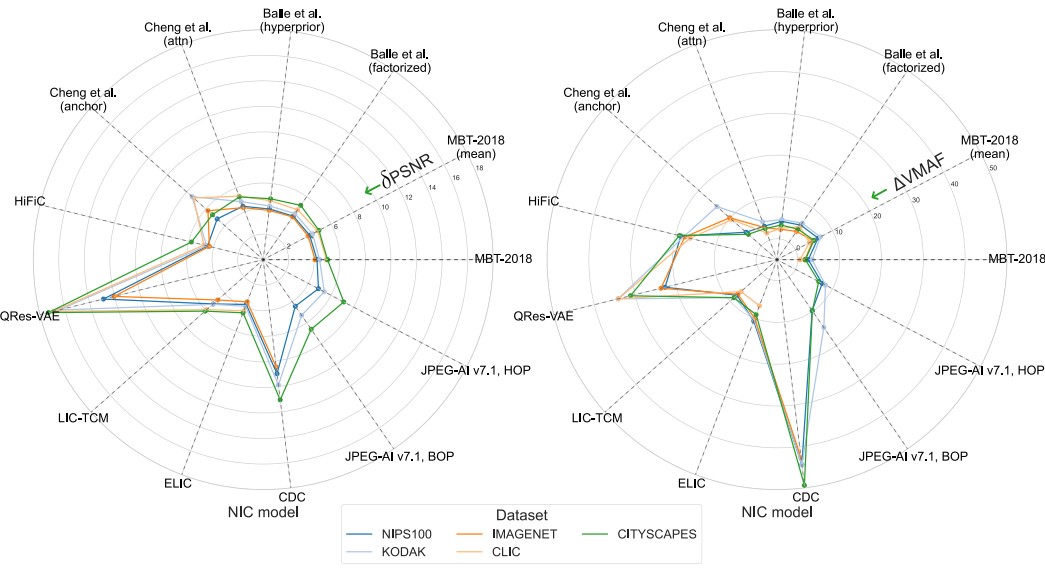

Figure 16: NIC performance across different datasets measured with $\delta PSNR \downarrow$ and $\Delta VMAF \downarrow$.

## A.8 ADDITIONAL RESULTS

In this section we provide some additional detailed results.

Table 4 demonstrates $\Delta$VMAF and $\delta$VMAF scores across different attack objectives, with scores being aggregated across all attacks and codec variants (i.e., different compression ratios of the same

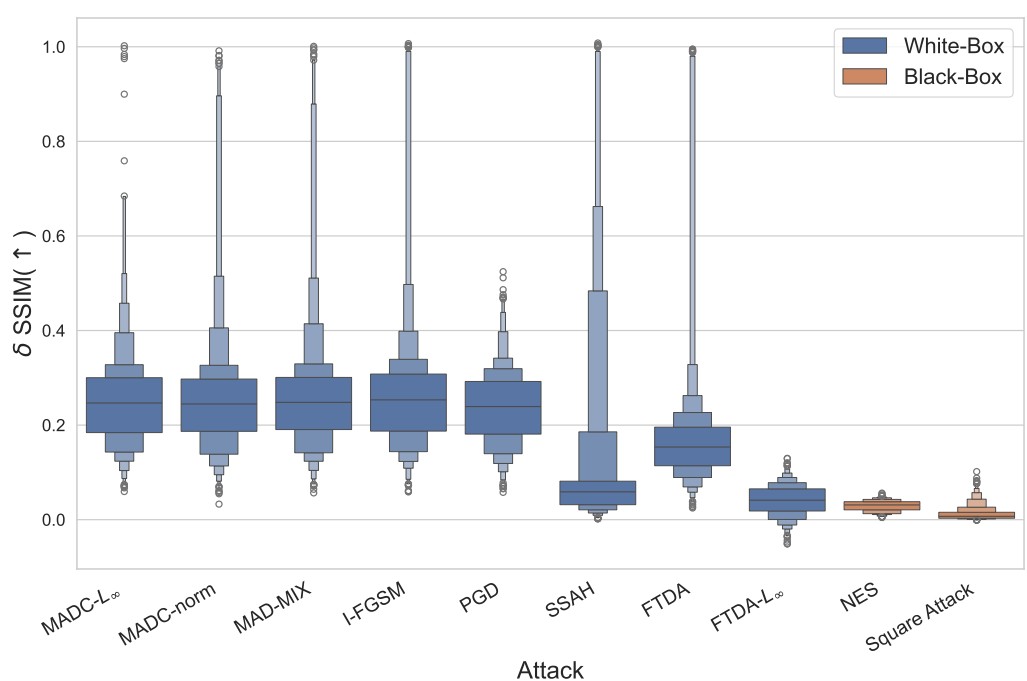

Figure 17: NIC performance across under different types of adversarial attacks measured with $\delta SSIM \uparrow$.

codec). Table 5 details these results and compares all NIC variants across different attacks objectives. Table 6 provides similar results for $\Delta$SSIM and $\delta$SSIM metrics.

Tables 7 and 8 compare NIC models across different adversarial attacks with the same optimization target.

Table 9 summarizes all evaluation scores across all attacks with "Reconstruction Loss" objective.

Figure 15 presents RD curves for SSIM across all attacks with "Reconstruction Loss" objective.

Figure 16 summarizes $\delta PSNR$ and $\Delta VMAF$ across different datasets with "Reconstruction Loss" objective.

Figure 17 compares different attacks and attack types with "Reconstruction Loss" objective.

| NIC | Rec. Loss | | BPP Loss | | FTDA Loss | | Added Noises | | Rec. Loss(Y) | | Source Rec.(Y) | |
|---|---|---|---|---|---|---|---|---|---|---|---|---|
| | $\Delta$VMAF↓ | $\delta$VMAF↓ | $\Delta$VMAF↓ | $\delta$VMAF↓ | $\Delta$VMAF↓ | $\delta$VMAF↓ | $\Delta$VMAF↓ | $\delta$VMAF↓ | $\Delta$VMAF↓ | $\delta$VMAF↓ | $\Delta$VMAF↓ | $\delta$VMAF↓ |
| MBT-2018 | **2.76** | **1.61** | 6.26 | 3.03 | **6.9** | 1.17 | **4.94** | 1.66 | 2.06 | 1.46 | 7.37 | 0.511 |
| Balle et al. (hyperprior) | 3.61 | 1.88 | 8.82 | 6.05 | 8.53 | 1.46 | 6.83 | 1.88 | 2.7 | 1.9 | 8.85 | 1.4 |
| Cheng et al. (attn) | 4.03 | 2.43 | 7.6 | 5.01 | 8.78 | 2.06 | 7.06 | 2.63 | 3.53 | 2.98 | 8.24 | 1.26 |
| Balle et al. (factorized) | 4.28 | 3.18 | 12.2 | 9.67 | 10.5 | 2.47 | 7.97 | 2.77 | 2.59 | 4.03 | 10.7 | 3.54 |
| MBT-2018 (mean) | 6.43 | 1.65 | 10.1 | 5.12 | 9.41 | 1.29 | 9.38 | 1.72 | 4.93 | **1.37** | 10.1 | 0.953 |
| JPEG-AI v7.1, HOP | 7.67 | 1.66 | 9.42 | **1.63** | 10.4 | 0.906 | 10.8 | **1.44** | 7.38 | 3 | **5.41** | 0.631 |
| LIC-TCM | 8.95 | 2.81 | **6.12** | 2.08 | 11.1 | 2.06 | 10.2 | 2.6 | 7.59 | 2.68 | 6.96 | 0.466 |
| JPEG-AI v7.1, BOP | 9.7 | 3.32 | 10.8 | 3.08 | 12 | 2.62 | 13.2 | 2.96 | 9.18 | 4 | 5.69 | 1.01 |
| ELIC | 11.9 | 2.82 | 13.1 | 1.66 | 16.5 | **-0.174** | 15.9 | 1.62 | 9.78 | 3.2 | 13.2 | **-0.372** |
| Cheng et al. (anchor) | 13.1 | 11.7 | 10.6 | 7.93 | 16 | 10.4 | 19.2 | 15.8 | 8.29 | 7.54 | 10.3 | 3.53 |
| QRes-VAE | 15.2 | 9.82 | nan | nan | 15.7 | 9.75 | 15.7 | 9.72 | 13.8 | 9.13 | 13.8 | 8.94 |
| HiFiC | 15.6 | 6.12 | 17.2 | 10.8 | 23.3 | 4.16 | 21.4 | 4.74 | 12.7 | 7.98 | 18.2 | 5.18 |
| CDC | 40.1 | 23.9 | 24.3 | 7.75 | 45.8 | 26.6 | 44.4 | 25.8 | 38 | 24.1 | 40.7 | 24.1 |

Table 4: $\Delta$VMAF (↓) and $\delta$VMAF (↓) scores for different NIC models across different attack objectives. Results are averaged across all attacks and all NIC variants.

| NIC | NIC Var. | Rec. Loss | | BPP Loss | | FTDA Loss | | Added Noises | | Rec. Loss(Y) | | Source Rec.(Y) | |
|---|---|---|---|---|---|---|---|---|---|---|---|---|---|
| | | ΔVMAF↓ | δVMAF↓ | ΔVMAF↓ | δVMAF↓ | ΔVMAF↓ | δVMAF↓ | ΔVMAF↓ | δVMAF↓ | ΔVMAF↓ | δVMAF↓ | ΔVMAF↓ | δVMAF↓ |
| MBT-2018 | 1 | 3.065 | 2.206 | 10.453 | 4.027 | 11.271 | 1.212 | 7.286 | 2.018 | 2.083 | 2.594 | 13.281 | 0.373 |
| | 2 | 2.881 | 1.934 | 7.999 | 4.780 | 8.513 | 1.117 | 5.701 | 1.772 | 1.741 | 1.857 | 9.674 | 0.728 |
| | 4 | 3.125 | 1.568 | 4.520 | 2.333 | 5.432 | 1.713 | 4.563 | 2.152 | 2.556 | 1.225 | 4.603 | 0.662 |
| | 6 | 1.982 | **0.715** | 2.049 | 0.995 | **2.394** | 0.626 | 2.225 | 0.707 | 1.846 | **0.168** | 1.915 | 0.282 |
| Balle et al.(hyperprior) | 1 | 4.254 | 2.515 | 11.623 | 4.906 | 13.746 | 1.621 | 11.127 | 2.625 | 2.502 | 2.881 | 15.033 | 1.773 |
| | 2 | 4.424 | 2.506 | 10.536 | 6.922 | 10.881 | 1.562 | 8.810 | 2.407 | 3.134 | 2.858 | 12.367 | 2.388 |
| | 4 | 3.197 | 1.493 | 6.276 | 4.295 | 5.886 | 1.271 | 4.547 | 1.453 | 2.612 | 1.411 | 5.477 | 0.894 |
| | 6 | 2.549 | 1.007 | 6.849 | 8.075 | 3.606 | 1.382 | 2.854 | 1.018 | 2.542 | 0.449 | 2.504 | 0.545 |
| Cheng et al.(attn) | 1 | 4.449 | 2.528 | 12.313 | 6.619 | 12.816 | 1.400 | 10.165 | 2.498 | 3.359 | 3.115 | 13.592 | 0.892 |
| | 2 | 4.144 | 2.427 | 9.986 | 6.699 | 10.571 | 1.807 | 8.709 | 2.896 | 3.038 | 2.917 | 10.354 | 0.893 |
| | 4 | 3.909 | 1.920 | 5.482 | 3.929 | 6.284 | 1.826 | 5.029 | 2.149 | 4.099 | 2.364 | 5.411 | 1.200 |
| | 6 | 3.631 | 2.829 | 2.614 | 2.804 | 5.465 | 3.210 | 4.334 | 2.972 | 3.624 | 3.532 | 3.619 | 2.066 |
| Balle et al.(factorized) | 1 | 5.864 | 4.107 | 17.571 | 9.509 | 16.675 | 2.639 | 13.566 | 3.787 | 3.529 | 5.361 | 17.635 | 4.493 |
| | 2 | 4.944 | 3.844 | 13.984 | 9.646 | 12.835 | 2.489 | 9.626 | 3.195 | 3.198 | 5.141 | 13.178 | 3.902 |
| | 4 | 3.621 | 2.820 | 9.906 | 9.827 | 8.006 | 2.573 | 5.481 | 2.290 | 1.801 | 3.428 | 7.373 | 2.691 |
| | 6 | 2.672 | 1.957 | 7.175 | 9.718 | 4.469 | 2.197 | 3.192 | 1.792 | 1.824 | 2.197 | 4.675 | 3.071 |
| MBT-2018(mean) | 1 | 8.533 | 2.138 | 15.155 | 5.821 | 14.578 | 1.242 | 14.473 | 2.205 | 6.035 | 2.257 | 16.876 | 1.059 |
| | 2 | 8.257 | 2.087 | 14.212 | 9.720 | 12.215 | 1.430 | 12.401 | 2.020 | 6.312 | 2.306 | 13.956 | 1.416 |
| | 4 | 5.007 | 1.056 | 7.789 | 4.053 | 6.903 | 1.144 | 6.447 | 1.219 | 4.015 | 0.673 | 6.723 | 0.842 |
| | 6 | 3.912 | 1.332 | 3.297 | 0.899 | 3.928 | 1.339 | 4.202 | 1.425 | 3.348 | 0.240 | 2.791 | 0.495 |
| JPEG-AI v7.1, HOP | b0002 | 16.943 | 2.509 | 20.544 | 3.137 | 20.577 | -0.029 | 23.113 | 1.777 | 14.836 | 3.065 | 8.500 | **-3.702** |
| | b0012 | 8.556 | 1.630 | 11.130 | 1.452 | 11.665 | 0.611 | 12.702 | 1.487 | 6.681 | 1.401 | 3.332 | -1.837 |
| | b0075 | 3.782 | 1.354 | 4.377 | 1.260 | 6.622 | 1.843 | 5.516 | 1.549 | 3.889 | 2.580 | 3.379 | 1.831 |
| | b05 | 1.403 | 1.155 | 1.623 | 0.674 | 2.709 | 1.199 | 1.947 | 0.945 | 4.108 | 4.943 | 6.427 | 6.231 |
| LIC-TCM | 00025 | 13.894 | 3.132 | 10.972 | 2.076 | 17.056 | 1.242 | 15.854 | 2.200 | 11.520 | 3.432 | 12.481 | -0.139 |
| | 0013 | 8.527 | 2.979 | 5.413 | 1.360 | 10.555 | 2.957 | 9.815 | 3.279 | 6.885 | 2.175 | 5.429 | 0.489 |
| | 005 | 4.428 | 2.308 | 1.970 | 2.804 | 5.772 | 1.982 | 4.792 | 2.320 | 4.370 | 2.441 | 2.955 | 1.049 |
| JPEG-AI v7.1, BOP | b0002 | 20.552 | 4.863 | 25.645 | 8.451 | 24.434 | 4.217 | 28.432 | 4.844 | 20.291 | 6.324 | 13.470 | 0.989 |
| | b0012 | 13.760 | 6.125 | 12.538 | 2.627 | 15.124 | 4.569 | 17.577 | 5.566 | 13.528 | 7.099 | 6.915 | 2.415 |
| | b0075 | 3.663 | 1.172 | 3.611 | **0.580** | 5.960 | 0.818 | 5.234 | 0.768 | 2.993 | 1.452 | 1.504 | -0.036 |
| | b05 | **0.821** | 1.129 | **1.597** | 0.648 | 2.670 | 0.872 | **1.622** | **0.648** | -0.087 | 1.125 | **0.887** | 0.665 |
| ELIC | 0004 | 17.238 | 3.481 | 22.848 | 2.008 | 26.179 | -0.798 | 26.395 | 1.850 | 12.869 | 4.273 | 22.722 | -0.709 |
| | 0016 | 14.652 | 3.341 | 14.193 | 2.210 | 18.371 | -0.357 | 17.630 | 1.792 | 12.656 | 3.909 | 14.618 | -0.559 |
| | 0450 | 3.928 | 1.644 | 2.405 | 0.770 | 4.833 | 0.633 | 3.751 | 1.232 | 3.812 | 1.421 | 2.338 | 0.151 |
| Cheng et al.(anchor) | 1 | 3.905 | 2.370 | 11.918 | 4.316 | 12.944 | 1.684 | 9.683 | 2.575 | 3.382 | 3.021 | 13.923 | 0.924 |
| | 2 | 3.600 | 2.165 | 8.780 | 4.265 | 10.379 | 1.913 | 7.726 | 2.579 | 3.303 | 2.986 | 10.287 | 1.068 |
| | 4 | 17.979 | 16.120 | 15.410 | 16.257 | 16.349 | 13.554 | 26.546 | 25.140 | 7.812 | 5.866 | 7.049 | 3.131 |
| | 6 | 26.720 | 26.246 | 6.397 | 6.880 | 24.504 | 24.471 | 32.668 | 33.086 | 18.670 | 18.269 | 9.921 | 9.012 |
| QRes-VAE | 16 | 17.371 | 1.671 | — | — | 15.627 | 0.366 | 18.154 | 0.890 | 15.494 | 1.666 | 14.770 | 1.914 |
| | 2048 | 24.717 | 26.577 | — | — | 28.075 | 28.358 | 25.605 | 27.424 | 22.489 | 24.698 | 24.006 | 24.354 |
| | 256 | 3.535 | 1.225 | — | — | 3.336 | 0.537 | 3.387 | 0.855 | 3.558 | 1.011 | 2.689 | 0.547 |
| HiFiC | 014 | 19.523 | 6.024 | 24.359 | 9.207 | 29.404 | 2.981 | 28.052 | 4.332 | 17.015 | 8.209 | 23.408 | 4.068 |
| | 030 | 10.233 | 6.281 | 11.389 | 12.022 | 17.778 | 5.329 | 15.070 | 5.074 | 6.790 | 7.647 | 12.680 | 5.284 |
| | 045 | 17.023 | 6.047 | 15.725 | 11.264 | 22.653 | 4.178 | 21.044 | 4.823 | 14.162 | 8.085 | 18.369 | 6.175 |
| CDC | b00032 | 17.675 | 15.742 | 5.305 | 4.852 | 19.952 | 15.799 | 17.841 | 13.982 | 11.755 | 12.624 | 12.842 | 10.172 |
| | b01024 | 43.778 | 28.238 | 27.451 | 10.888 | 52.758 | 32.846 | 51.538 | 32.421 | 44.140 | 31.559 | 48.557 | 33.384 |
| | b02048 | 58.986 | 27.855 | 40.256 | 7.498 | 64.578 | 31.155 | 63.887 | 31.027 | 58.227 | 28.248 | 60.809 | 28.636 |

Table 5: ΔVMAF (↓) and δVMAF (↓) scores for different NIC models and their variants across different attack objectives. Results are averaged across all attacks.

| NIC | NIC Var. | Rec. Loss | | BPP Loss | | FTDA Loss | | Added Noises | | Rec. Loss(Y) | | Source Rec.(Y) | |
|---|---|---|---|---|---|---|---|---|---|---|---|---|---|
| | | ΔSSIM↓ | δSSIM↓ | ΔSSIM↓ | δSSIM↓ | ΔSSIM↓ | δSSIM↓ | ΔSSIM↓ | δSSIM↓ | ΔSSIM↓ | δSSIM↓ | ΔSSIM↓ | δSSIM↓ |
| Balle et al.(hyperprior) | 1 | -0.116 | 0.154 | -0.034 | 0.094 | -0.007 | 0.036 | -0.105 | 0.110 | -0.141 | 0.195 | -0.011 | 0.077 |
| | 2 | -0.117 | 0.154 | -0.018 | 0.074 | -0.006 | 0.033 | -0.106 | 0.111 | -0.136 | 0.191 | -0.015 | 0.081 |
| | 4 | -0.120 | 0.146 | -0.020 | 0.057 | 0.001 | 0.025 | -0.113 | 0.119 | -0.132 | 0.168 | -0.012 | 0.046 |
| | 6 | -0.113 | **0.133** | -0.011 | 0.067 | 0.004 | 0.032 | -0.110 | 0.124 | -0.109 | 0.123 | -0.010 | 0.024 |
| Balle et al.(factorized) | 1 | -0.112 | 0.162 | -0.020 | 0.122 | **-0.021** | 0.056 | -0.109 | 0.123 | **-0.142** | 0.206 | -0.020 | 0.080 |
| | 2 | -0.113 | 0.161 | -0.007 | 0.107 | -0.014 | 0.046 | -0.109 | 0.121 | -0.139 | 0.206 | -0.015 | 0.066 |
| | 4 | -0.116 | 0.157 | 0.002 | 0.081 | -0.006 | 0.036 | -0.112 | 0.123 | -0.136 | 0.197 | -0.012 | 0.044 |
| | 6 | -0.116 | 0.145 | 0.005 | 0.068 | -0.003 | 0.032 | -0.113 | 0.127 | -0.129 | 0.167 | -0.012 | 0.031 |
| MBT-2018 | 1 | -0.114 | 0.150 | -0.064 | 0.115 | 0.015 | 0.022 | -0.102 | 0.109 | -0.128 | 0.189 | 0.012 | 0.052 |
| | 2 | -0.119 | 0.149 | -0.043 | 0.088 | 0.005 | 0.023 | -0.111 | 0.114 | -0.133 | 0.185 | -0.006 | 0.053 |
| | 4 | -0.113 | 0.146 | -0.034 | 0.072 | 0.010 | 0.039 | -0.104 | 0.129 | -0.118 | 0.163 | -0.001 | 0.036 |
| | 6 | -0.106 | 0.139 | -0.030 | 0.052 | 0.002 | 0.035 | -0.104 | 0.130 | -0.101 | 0.123 | -0.008 | 0.020 |
| MBT-2018(mean) | 1 | -0.114 | 0.155 | -0.037 | 0.103 | 0.006 | 0.026 | -0.100 | **0.109** | -0.139 | 0.197 | 0.000 | 0.071 |
| | 2 | -0.115 | 0.157 | -0.010 | 0.092 | 0.007 | 0.025 | -0.104 | 0.112 | -0.135 | 0.195 | -0.003 | 0.075 |
| | 4 | -0.119 | 0.146 | -0.014 | 0.053 | -0.002 | 0.031 | -0.113 | 0.122 | -0.129 | 0.169 | -0.015 | 0.049 |
| | 6 | -0.094 | 0.145 | -0.016 | 0.064 | 0.005 | 0.057 | -0.089 | 0.137 | -0.102 | 0.127 | -0.008 | 0.022 |
| JPEG-AI v7.1, BOP | b0002 | -0.095 | 0.186 | 0.101 | 0.056 | 0.044 | 0.063 | -0.088 | 0.150 | -0.085 | 0.188 | **-0.045** | 0.052 |
| | b0012 | -0.102 | 0.207 | 0.056 | 0.031 | 0.043 | 0.046 | -0.111 | 0.175 | -0.091 | 0.199 | -0.019 | 0.028 |
| | b0075 | -0.116 | 0.203 | 0.017 | 0.034 | 0.029 | 0.026 | -0.119 | 0.182 | -0.114 | 0.171 | -0.010 | **0.008** |
| | b05 | -0.115 | 0.203 | -0.003 | 0.036 | 0.024 | 0.035 | -0.115 | 0.186 | -0.129 | 0.179 | -0.004 | 0.013 |
| Cheng et al.(attn) | 1 | -0.112 | 0.154 | 0.016 | 0.075 | 0.007 | 0.034 | -0.099 | 0.114 | -0.125 | 0.192 | 0.010 | 0.060 |
| | 2 | -0.112 | 0.154 | 0.014 | 0.065 | 0.004 | 0.036 | -0.101 | 0.119 | -0.124 | 0.192 | 0.007 | 0.056 |
| | 4 | -0.110 | 0.149 | 0.005 | 0.045 | 0.004 | 0.036 | -0.105 | 0.130 | -0.104 | 0.166 | -0.002 | 0.036 |
| | 6 | -0.090 | 0.155 | -0.011 | 0.047 | 0.008 | 0.053 | -0.087 | 0.144 | -0.057 | 0.182 | -0.006 | 0.033 |
| ELIC | 0004 | -0.085 | 0.156 | -0.074 | 0.118 | 0.030 | 0.030 | -0.079 | 0.110 | -0.115 | 0.194 | 0.004 | 0.068 |
| | 0016 | -0.089 | 0.156 | **-0.077** | 0.119 | 0.030 | 0.022 | -0.088 | 0.113 | -0.114 | 0.195 | 0.013 | 0.051 |
| | 0450 | -0.107 | 0.141 | -0.051 | 0.069 | 0.007 | 0.023 | -0.108 | 0.128 | -0.082 | **0.107** | -0.011 | 0.020 |
| LIC-TCM | 00025 | -0.087 | 0.156 | -0.066 | 0.109 | 0.031 | 0.022 | -0.087 | 0.115 | -0.117 | 0.196 | 0.020 | 0.047 |
| | 0013 | -0.083 | 0.153 | -0.070 | 0.103 | 0.025 | 0.058 | -0.074 | 0.137 | -0.111 | 0.164 | -0.006 | 0.031 |
| | 005 | -0.081 | 0.153 | -0.074 | 0.101 | 0.019 | 0.057 | -0.076 | 0.147 | -0.089 | 0.125 | -0.014 | 0.025 |
| Cheng et al.(anchor) | 1 | -0.110 | 0.154 | -0.008 | 0.077 | 0.010 | 0.034 | -0.097 | 0.116 | -0.118 | 0.192 | 0.013 | 0.061 |
| | 2 | -0.110 | 0.152 | 0.003 | 0.061 | 0.010 | 0.034 | -0.097 | 0.118 | -0.112 | 0.192 | 0.010 | 0.054 |
| | 4 | -0.048 | 0.174 | 0.047 | 0.104 | 0.037 | 0.077 | 0.000 | 0.187 | -0.091 | 0.171 | 0.006 | 0.041 |
| | 6 | -0.008 | 0.186 | -0.003 | 0.055 | 0.053 | 0.124 | 0.028 | 0.207 | -0.031 | 0.183 | 0.005 | 0.049 |
| JPEG-AI v7.1, HOP | b0002 | -0.099 | 0.185 | 0.075 | 0.033 | 0.049 | 0.040 | -0.095 | 0.145 | -0.107 | 0.198 | -0.014 | 0.011 |
| | b0012 | **-0.125** | 0.209 | 0.054 | **0.023** | 0.039 | 0.041 | **-0.123** | 0.169 | -0.127 | 0.209 | -0.016 | 0.014 |
| | b0075 | -0.111 | 0.223 | 0.051 | 0.035 | 0.097 | 0.112 | -0.096 | 0.200 | -0.084 | 0.219 | 0.018 | 0.028 |
| | b05 | 0.071 | 0.302 | 0.016 | 0.044 | 0.104 | 0.138 | 0.032 | 0.253 | 0.001 | 0.227 | 0.067 | 0.084 |
| QRes-VAE | 16 | -0.084 | 0.147 | — | — | 0.044 | **0.010** | -0.076 | 0.113 | -0.110 | 0.180 | 0.031 | 0.039 |
| | 2048 | 0.183 | 0.337 | — | — | 0.245 | 0.303 | 0.185 | 0.344 | 0.180 | 0.313 | 0.199 | 0.256 |
| | 256 | -0.108 | 0.136 | — | — | 0.007 | 0.023 | -0.105 | 0.125 | -0.088 | 0.113 | -0.008 | 0.022 |
| HiFiC | 014 | 0.065 | 0.250 | 0.049 | 0.133 | 0.120 | 0.114 | 0.095 | 0.206 | 0.045 | 0.289 | 0.063 | 0.159 |
| | 030 | 0.024 | 0.260 | 0.029 | 0.103 | 0.091 | 0.125 | 0.048 | 0.216 | 0.018 | 0.316 | 0.061 | 0.170 |
| | 045 | 0.050 | 0.269 | 0.035 | 0.088 | 0.112 | 0.135 | 0.070 | 0.230 | 0.038 | 0.306 | 0.058 | 0.156 |
| CDC | b00032 | 0.111 | 0.271 | 0.007 | 0.076 | 0.165 | 0.225 | 0.105 | 0.243 | 0.094 | 0.279 | 0.088 | 0.165 |
| | b01024 | 0.104 | 0.270 | 0.077 | 0.124 | 0.126 | 0.202 | 0.107 | 0.233 | 0.091 | 0.284 | 0.100 | 0.235 |
| | b02048 | 0.152 | 0.283 | 0.076 | 0.097 | 0.168 | 0.260 | 0.159 | 0.266 | 0.151 | 0.290 | 0.156 | 0.278 |

Table 6: ΔSSIM (↓) and δSSIM (↓) scores for different NIC models and their variants across different attack objectives. Results are averaged across all attacks.

| NIC | NIC Var. | FTDA | FTDA-$L_\infty$ | I-FGSM | MAD-MIX | MADC | MADC-$L_\infty$ | MADC-norm | PGD | SSAH |
|---|---|---|---|---|---|---|---|---|---|---|
| Balle et al.(hyperprior) | 1 | -0.094 ±0.048 | -0.068 ±0.032 | -0.155 ±0.064 | -0.164 ±0.068 | 0.001 ±0.000 | -0.163 ±0.068 | -0.165 ±0.069 | -0.176 ±0.072 | -0.087 ±0.043 |
| | 2 | -0.092 ±0.043 | -0.067 ±0.032 | -0.161 ±0.065 | -0.170 ±0.069 | 0.001 ±0.000 | -0.169 ±0.068 | -0.171 ±0.069 | -0.181 ±0.072 | -0.074 ±0.038 |
| | 4 | -0.105 ±0.053 | -0.068 ±0.034 | -0.178 ±0.068 | **-0.183** ±0.070 | **0.001** ±0.000 | **-0.182** ±0.069 | **-0.183** ±0.070 | **-0.189** ±0.072 | -0.047 ±0.027 |
| | 6 | -0.090 ±0.049 | -0.067 ±0.036 | **-0.179** ±0.070 | -0.179 ±0.070 | 0.001 ±0.000 | -0.177 ±0.069 | -0.178 ±0.070 | -0.183 ±0.071 | -0.022 ±0.018 |
| MBT-2018 | 1 | -0.101 ±0.054 | -0.068 ±0.031 | -0.150 ±0.061 | -0.166 ±0.068 | 0.001 ±0.000 | -0.165 ±0.067 | -0.168 ±0.069 | -0.177 ±0.072 | -0.061 ±0.034 |
| | 2 | -0.109 ±0.053 | -0.068 ±0.032 | -0.163 ±0.062 | -0.175 ±0.068 | 0.001 ±0.000 | -0.174 ±0.067 | -0.176 ±0.069 | -0.185 ±0.071 | -0.061 ±0.033 |
| | 4 | -0.086 ±0.066 | -0.069 ±0.036 | -0.170 ±0.068 | -0.176 ±0.071 | 0.001 ±0.000 | -0.175 ±0.070 | -0.177 ±0.071 | -0.184 ±0.072 | -0.032 ±0.029 |
| | 6 | -0.073 ±0.057 | -0.069 ±0.036 | -0.162 ±0.073 | -0.171 ±0.070 | 0.001 ±0.001 | -0.169 ±0.069 | -0.172 ±0.069 | -0.176 ±0.070 | -0.017 ±0.032 |
| Balle et al.(factorized) | 1 | -0.090 ±0.062 | -0.065 ±0.033 | -0.147 ±0.068 | -0.154 ±0.071 | 0.001 ±0.000 | -0.153 ±0.071 | -0.155 ±0.072 | -0.166 ±0.075 | -0.092 ±0.047 |
| | 2 | -0.088 ±0.058 | -0.065 ±0.033 | -0.152 ±0.068 | -0.161 ±0.072 | 0.001 ±0.000 | -0.160 ±0.072 | -0.162 ±0.073 | -0.173 ±0.075 | -0.075 ±0.039 |
| | 4 | -0.092 ±0.061 | -0.066 ±0.034 | -0.165 ±0.069 | -0.173 ±0.072 | 0.001 ±0.000 | -0.172 ±0.072 | -0.174 ±0.073 | -0.183 ±0.074 | -0.053 ±0.031 |
| | 6 | -0.080 ±0.053 | -0.068 ±0.036 | -0.177 ±0.070 | -0.181 ±0.071 | 0.001 ±0.000 | -0.180 ±0.071 | -0.181 ±0.072 | -0.189 ±0.072 | -0.031 ±0.024 |
| JPEG-AI v7.1, BOP | b0002 | -0.066 ±0.061 | -0.074 ±0.052 | -0.108 ±0.061 | -0.109 ±0.065 | 0.004 ±0.003 | -0.111 ±0.066 | -0.123 ±0.069 | -0.124 ±0.071 | -0.126 ±0.082 |
| | b0012 | -0.080 ±0.063 | -0.080 ±0.054 | -0.104 ±0.059 | -0.106 ±0.065 | 0.003 ±0.001 | -0.098 ±0.064 | -0.128 ±0.061 | -0.135 ±0.069 | -0.143 ±0.082 |
| | b0075 | -0.106 ±0.048 | -0.087 ±0.038 | -0.129 ±0.055 | -0.146 ±0.063 | 0.002 ±0.001 | -0.137 ±0.059 | -0.149 ±0.065 | -0.148 ±0.064 | -0.152 ±0.043 |
| | b05 | -0.097 ±0.034 | -0.101 ±0.035 | -0.117 ±0.040 | -0.125 ±0.048 | 0.001 ±0.000 | -0.121 ±0.047 | -0.129 ±0.048 | -0.140 ±0.049 | -0.185 ±0.051 |
| Cheng et al.(attn) | 1 | -0.105 ±0.055 | -0.076 ±0.036 | -0.145 ±0.066 | -0.157 ±0.072 | 0.001 ±0.000 | -0.156 ±0.072 | -0.159 ±0.073 | -0.170 ±0.075 | -0.063 ±0.037 |
| | 2 | -0.101 ±0.054 | -0.074 ±0.037 | -0.151 ±0.068 | -0.160 ±0.073 | 0.001 ±0.002 | -0.159 ±0.073 | -0.163 ±0.074 | -0.174 ±0.076 | -0.054 ±0.033 |
| | 4 | -0.091 ±0.057 | -0.070 ±0.038 | -0.162 ±0.072 | -0.165 ±0.076 | 0.001 ±0.002 | -0.163 ±0.076 | -0.168 ±0.075 | -0.177 ±0.077 | -0.032 ±0.025 |
| | 6 | -0.035 ±0.052 | -0.067 ±0.037 | -0.143 ±0.078 | -0.128 ±0.087 | 0.001 ±0.002 | -0.120 ±0.088 | -0.141 ±0.084 | -0.160 ±0.082 | -0.027 ±0.025 |
| MBT-2018(mean) | 1 | -0.096 ±0.054 | -0.066 ±0.032 | -0.150 ±0.063 | -0.161 ±0.069 | 0.001 ±0.001 | -0.160 ±0.068 | -0.163 ±0.069 | -0.173 ±0.072 | -0.080 ±0.041 |
| | 2 | -0.100 ±0.053 | -0.065 ±0.032 | -0.157 ±0.064 | -0.165 ±0.068 | 0.001 ±0.001 | -0.165 ±0.068 | -0.167 ±0.069 | -0.178 ±0.072 | -0.071 ±0.038 |
| | 4 | -0.106 ±0.054 | -0.067 ±0.034 | -0.176 ±0.067 | -0.181 ±0.069 | 0.001 ±0.001 | -0.181 ±0.069 | -0.182 ±0.070 | -0.188 ±0.071 | -0.044 ±0.027 |
| | 6 | -0.013 ±0.048 | -0.069 ±0.036 | -0.152 ±0.078 | -0.147 ±0.080 | 0.001 ±0.001 | -0.151 ±0.077 | -0.152 ±0.077 | -0.172 ±0.075 | -0.021 ±0.019 |
| JPEG-AI v7.1, HOP | b0002 | -0.079 ±0.053 | -0.069 ±0.053 | -0.111 ±0.063 | -0.114 ±0.072 | 0.005 ±0.003 | -0.111 ±0.067 | -0.117 ±0.072 | -0.126 ±0.077 | -0.139 ±0.078 |
| | b0012 | -0.100 ±0.053 | -0.099 ±0.055 | -0.126 ±0.063 | -0.138 ±0.073 | 0.003 ±0.001 | -0.136 ±0.074 | -0.141 ±0.073 | -0.152 ±0.075 | -0.189 ±0.085 |
| | b0075 | **-0.110** ±0.070 | **-0.117** ±0.064 | -0.127 ±0.091 | -0.060 ±0.182 | 0.002 ±0.001 | -0.067 ±0.166 | -0.081 ±0.157 | -0.163 ±0.081 | **-0.197** ±0.105 |
| | b05 | -0.018 ±0.095 | -0.075 ±0.054 | 0.047 ±0.124 | 0.342 ±0.162 | 0.001 ±0.000 | 0.383 ±0.147 | 0.273 ±0.205 | 0.043 ±0.085 | -0.087 ±0.097 |
| ELIC | 0004 | -0.052 ±0.069 | -0.040 ±0.046 | -0.098 ±0.075 | -0.112 ±0.084 | 0.008 ±0.007 | -0.111 ±0.083 | -0.113 ±0.084 | -0.124 ±0.088 | -0.094 ±0.058 |
| | 0016 | -0.059 ±0.060 | -0.051 ±0.042 | -0.116 ±0.071 | -0.132 ±0.079 | 0.010 ±0.006 | -0.130 ±0.079 | -0.133 ±0.080 | -0.144 ±0.083 | -0.050 ±0.038 |
| | 0450 | -0.071 ±0.049 | -0.066 ±0.036 | -0.166 ±0.070 | -0.171 ±0.075 | 0.004 ±0.002 | -0.168 ±0.074 | -0.173 ±0.073 | -0.183 ±0.074 | -0.015 ±0.018 |
| Cheng et al.(anchor) | 1 | -0.109 ±0.049 | -0.077 ±0.035 | -0.145 ±0.066 | -0.156 ±0.073 | 0.001 ±0.000 | -0.155 ±0.073 | -0.158 ±0.073 | -0.170 ±0.076 | -0.055 ±0.034 |
| | 2 | -0.103 ±0.061 | -0.075 ±0.036 | -0.148 ±0.069 | -0.158 ±0.076 | 0.001 ±0.000 | -0.156 ±0.076 | -0.161 ±0.076 | -0.172 ±0.077 | -0.045 ±0.030 |
| | 4 | -0.011 ±0.120 | -0.057 ±0.068 | -0.059 ±0.169 | -0.056 ±0.183 | 0.001 ±0.002 | -0.009 ±0.224 | -0.046 ±0.150 | -0.168 ±0.082 | 0.014 ±0.098 |
| | 6 | 0.027 ±0.096 | -0.058 ±0.034 | 0.032 ±0.169 | 0.054 ±0.179 | 0.002 ±0.003 | 0.085 ±0.177 | 0.015 ±0.149 | -0.151 ±0.089 | 0.023 ±0.087 |
| LIC-TCM | 00025 | -0.052 ±0.066 | -0.047 ±0.040 | -0.117 ±0.071 | -0.134 ±0.080 | 0.009 ±0.005 | -0.132 ±0.079 | -0.136 ±0.080 | -0.146 ±0.083 | -0.034 ±0.033 |
| | 0013 | 0.024 ±0.070 | -0.065 ±0.037 | -0.144 ±0.079 | -0.128 ±0.088 | 0.006 ±0.002 | -0.120 ±0.091 | -0.138 ±0.087 | -0.161 ±0.081 | -0.020 ±0.024 |
| | 005 | 0.007 ±0.053 | -0.066 ±0.035 | -0.134 ±0.066 | -0.117 ±0.084 | 0.003 ±0.001 | -0.117 ±0.084 | -0.132 ±0.080 | -0.165 ±0.077 | -0.013 ±0.017 |
| HiFiC | 014 | 0.138 ±0.048 | 0.004 ±0.040 | 0.094 ±0.044 | 0.119 ±0.046 | 0.008 ±0.006 | 0.125 ±0.047 | 0.103 ±0.044 | 0.089 ±0.046 | 0.048 ±0.049 |
| | 030 | 0.091 ±0.044 | -0.028 ±0.032 | 0.046 ±0.041 | 0.062 ±0.041 | 0.004 ±0.002 | 0.065 ±0.040 | 0.051 ±0.041 | 0.040 ±0.044 | 0.007 ±0.030 |
| | 045 | 0.119 ±0.041 | -0.019 ±0.031 | 0.081 ±0.043 | 0.105 ±0.046 | 0.006 ±0.003 | 0.109 ±0.045 | 0.090 ±0.046 | 0.074 ±0.047 | 0.027 ±0.029 |
| QRes-VAE | 16 | -0.072 ±0.058 | -0.040 ±0.045 | -0.123 ±0.073 | -0.136 ±0.079 | 0.023 ±0.016 | -0.134 ±0.078 | -0.146 ±0.080 | -0.013 ±0.031 | -0.013 ±0.031 |
| | 2048 | 0.718 ±0.367 | -0.020 ±0.131 | 0.595 ±0.429 | 0.293 ±0.450 | 0.065 ±0.201 | 0.042 ±0.292 | 0.349 ±0.467 | -0.113 ±0.047 | 0.111 ±0.254 |
| | 256 | -0.079 ±0.051 | -0.064 ±0.035 | -0.178 ±0.072 | -0.176 ±0.075 | 0.010 ±0.004 | -0.175 ±0.075 | -0.178 ±0.073 | -0.187 ±0.072 | -0.006 ±0.016 |
| CDC | b00032 | 0.202 ±0.113 | -0.055 ±0.041 | 0.225 ±0.156 | 0.288 ±0.158 | 0.007 ±0.005 | 0.299 ±0.152 | 0.248 ±0.169 | 0.090 ±0.165 | 0.028 ±0.079 |
| | b01024 | 0.160 ±0.055 | -0.005 ±0.042 | 0.148 ±0.057 | 0.153 ±0.064 | 0.022 ±0.014 | 0.152 ±0.066 | 0.152 ±0.061 | 0.122 ±0.060 | 0.119 ±0.049 |
| | b02048 | 0.216 ±0.060 | 0.029 ±0.064 | 0.218 ±0.072 | 0.210 ±0.073 | 0.047 ±0.029 | 0.206 ±0.072 | 0.216 ±0.071 | 0.194 ±0.072 | 0.148 ±0.085 |

Table 7: ΔSSIM (↓) scores for different NIC models and their variants across different adversarial attacks. "Reconstruction Loss" objective is used for all attacks.

| NIC | NIC Var. | FTDA | FTDA-$L_\infty$ | I-FGSM | MAD-MIX | MADC | MADC-$L_\infty$ | MADC-norm | PGD | SSAH |
|---|---|---|---|---|---|---|---|---|---|---|
| MBT-2018 | 1 | -4.770 ±2.040 | -3.131 ±1.817 | -5.168 ±1.518 | -5.773 ±1.755 | 13.970 ±2.596 | -5.737 ±1.731 | -5.867 ±1.840 | -6.320 ±1.938 | -5.470 ±2.945 |
| | 2 | -5.514 ±2.333 | -3.676 ±1.922 | -6.080 ±1.711 | -6.753 ±1.996 | 13.335 ±2.347 | -6.727 ±1.957 | -6.859 ±2.068 | -7.296 ±2.137 | -5.731 ±3.260 |
| | 4 | -2.502 ±3.775 | -3.336 ±2.385 | -5.823 ±2.242 | -6.370 ±2.514 | 13.316 ±2.480 | -6.325 ±2.431 | -6.463 ±2.487 | -6.667 ±2.387 | -2.632 ±4.348 |
| | 6 | -1.486 ±3.923 | -3.499 ±2.346 | -5.523 ±2.540 | -6.136 ±2.542 | 13.403 ±3.033 | -6.056 ±2.450 | -6.205 ±2.443 | -6.284 ±2.353 | -1.314 ±4.011 |
| Balle et al.(hyperprior) | 1 | -3.098 ±1.549 | -1.740 ±1.427 | -3.702 ±1.252 | -4.139 ±1.472 | 13.925 ±2.713 | -4.092 ±1.440 | -4.305 ±1.519 | -5.086 ±1.755 | -5.939 ±3.152 |
| | 2 | -3.394 ±1.768 | -2.426 ±1.605 | -4.608 ±1.366 | -5.100 ±1.609 | 13.989 ±2.292 | -5.116 ±1.585 | -5.275 ±1.643 | -6.010 ±1.887 | -5.531 ±3.298 |
| | 4 | -4.025 ±3.633 | -3.061 ±2.044 | -6.177 ±2.019 | -6.706 ±2.233 | 13.134 ±2.247 | -6.707 ±2.241 | -6.755 ±2.289 | -7.066 ±2.217 | -4.291 ±3.592 |
| | 6 | -2.616 ±5.389 | -3.035 ±2.418 | -6.147 ±2.360 | -6.321 ±2.595 | 13.327 ±2.815 | -6.246 ±2.522 | -6.358 ±2.526 | -6.509 ±2.304 | -1.490 ±4.308 |
| Cheng et al.(attn) | 1 | -3.681 ±2.052 | -2.747 ±1.980 | -4.072 ±1.879 | -4.322 ±2.143 | 14.633 ±2.850 | -4.236 ±2.135 | -4.476 ±2.124 | -5.265 ±2.280 | -4.036 ±3.111 |
| | 2 | -3.170 ±2.326 | -2.711 ±2.269 | -4.178 ±2.101 | -4.407 ±2.421 | 14.752 ±3.097 | -4.332 ±2.420 | -4.599 ±2.401 | -5.381 ±2.525 | -3.457 ±3.184 |
| | 4 | -2.093 ±3.266 | -3.020 ±2.581 | -4.066 ±2.267 | -4.233 ±2.760 | 14.794 ±3.630 | -4.066 ±2.786 | -4.521 ±2.634 | -5.185 ±2.686 | -1.569 ±3.355 |
| | 6 | 2.450 ±2.483 | -2.763 ±2.685 | -2.606 ±2.685 | -1.374 ±3.055 | 15.685 ±4.290 | -0.967 ±2.987 | -2.054 ±3.047 | -3.712 ±3.041 | -1.082 ±3.683 |
| JPEG-AI v7.1, BOP | b0002 | 0.704 ±2.846 | 0.416 ±2.261 | -0.453 ±1.987 | -0.158 ±1.986 | 22.008 ±3.562 | -0.168 ±2.017 | -0.734 ±1.835 | -0.580 ±1.899 | -0.386 ±3.707 |
| | b0012 | -0.814 ±3.265 | -0.894 ±2.225 | -1.300 ±2.983 | -1.080 ±2.985 | 20.462 ±2.029 | -0.190 ±4.015 | -2.156 ±1.853 | -2.038 ±2.382 | -1.849 ±3.620 |
| | b0075 | -2.966 ±1.163 | -2.329 ±0.795 | -3.401 ±1.264 | -3.861 ±1.519 | 17.935 ±1.577 | -3.553 ±1.429 | -4.031 ±1.568 | -3.875 ±1.422 | -3.777 ±0.708 |
| | b05 | -2.015 ±1.117 | -2.562 ±0.983 | -1.641 ±1.225 | -0.890 ±1.434 | 15.006 ±1.390 | -0.590 ±1.447 | -1.417 ±1.367 | -3.872 ±1.311 | -3.019 ±1.316 |
| MBT-2018(mean) | 1 | -2.005 ±1.806 | -1.206 ±1.511 | -3.733 ±1.533 | -3.881 ±1.580 | 17.098 ±3.868 | -3.885 ±1.598 | -4.021 ±1.600 | -4.559 ±1.738 | -4.839 ±4.074 |
| | 2 | -2.333 ±1.754 | -1.525 ±1.509 | -4.369 ±1.508 | -4.564 ±1.624 | 16.813 ±3.465 | -4.605 ±1.615 | -4.705 ±1.661 | -5.299 ±1.807 | -4.434 ±4.258 |
| | 4 | -4.100 ±3.036 | -2.533 ±2.008 | -6.000 ±1.950 | -6.437 ±2.127 | 16.615 ±2.250 | -6.460 ±2.121 | -6.358 ±2.118 | -6.738 ±2.109 | -3.242 ±4.281 |
| | 6 | 3.905 ±2.287 | -3.079 ±2.319 | -4.203 ±3.444 | -3.785 ±3.787 | 16.757 ±2.572 | -4.077 ±3.689 | -4.089 ±3.655 | -5.339 ±3.313 | 0.269 ±4.509 |
| JPEG-AI v7.1, HOP | b0002 | 0.021 ±1.626 | -0.418 ±1.676 | -0.364 ±1.804 | -0.364 ±1.804 | 22.125 ±8.460 | -0.310 ±1.795 | -0.444 ±1.808 | -0.641 ±1.828 | -1.580 ±1.508 |
| | b0012 | -1.843 ±1.632 | -1.711 ±1.612 | -2.410 ±1.710 | -2.409 ±2.033 | 20.391 ±2.375 | -2.336 ±2.055 | -2.495 ±1.967 | -2.832 ±1.928 | -3.768 ±1.793 |
| | b0075 | -2.523 ±2.456 | -2.935 ±1.881 | -2.646 ±3.166 | -0.474 ±6.352 | 18.015 ±1.711 | -0.733 ±5.771 | -1.280 ±5.374 | -4.417 ±2.582 | -3.905 ±2.470 |
| | b05 | 0.442 ±2.942 | -1.767 ±1.850 | 2.355 ±3.587 | 10.824 ±3.740 | 14.980 ±1.417 | 11.076 ±3.025 | 8.933 ±5.092 | 2.276 ±2.728 | -1.238 ±2.448 |
| Balle et al.(factorized) | 1 | -0.176 ±2.345 | -0.992 ±1.583 | -2.414 ±1.522 | -2.540 ±1.742 | 14.386 ±2.891 | -2.492 ±1.745 | -2.741 ±1.852 | -3.542 ±1.997 | -4.854 ±3.309 |
| | 2 | -0.671 ±2.939 | -2.004 ±1.757 | -3.431 ±1.606 | -3.763 ±1.993 | 14.061 ±2.274 | -3.679 ±1.967 | -3.948 ±1.993 | -4.689 ±2.170 | -5.162 ±3.347 |
| | 4 | -0.967 ±4.154 | -2.899 ±1.987 | -4.887 ±1.970 | -5.238 ±2.346 | 13.489 ±2.016 | -5.189 ±2.341 | -5.423 ±2.362 | -6.062 ±2.343 | -4.621 ±3.377 |
| | 6 | 1.155 ±4.534 | -3.314 ±2.450 | -6.027 ±2.429 | -6.374 ±2.859 | 12.879 ±2.384 | -6.300 ±2.879 | -6.488 ±2.939 | -7.005 ±2.604 | -2.945 ±4.086 |
| ELIC | 0004 | 1.759 ±2.292 | 3.633 ±2.043 | -0.037 ±1.982 | 0.047 ±2.046 | 22.693 ±9.488 | 0.069 ±2.033 | 0.041 ±2.047 | -0.190 ±2.088 | -3.410 ±5.191 |
| | 0016 | 0.323 ±2.272 | 1.803 ±1.940 | -1.591 ±1.927 | -1.596 ±1.990 | 25.496 ±4.463 | -1.553 ±1.979 | -1.652 ±2.026 | -1.873 ±2.067 | -1.764 ±4.972 |
| | 0450 | -1.831 ±2.316 | -2.774 ±2.010 | -4.980 ±2.082 | -5.045 ±2.564 | 21.872 ±2.461 | -4.832 ±2.562 | -5.329 ±2.463 | -6.128 ±2.358 | 0.522 ±4.924 |
| Cheng et al.(anchor) | 1 | -3.634 ±2.289 | -2.666 ±2.008 | -3.699 ±1.971 | -3.745 ±2.186 | 14.939 ±2.432 | -3.728 ±2.209 | -3.909 ±2.170 | -4.767 ±2.316 | -3.451 ±2.952 |
| | 2 | -2.724 ±2.706 | -2.750 ±1.995 | -3.581 ±2.160 | -3.721 ±2.482 | 14.670 ±2.381 | -3.618 ±2.495 | -3.964 ±2.458 | -4.807 ±2.513 | -2.731 ±3.009 |
| | 4 | 5.835 ±10.508 | -1.297 ±5.636 | 3.989 ±11.583 | 3.640 ±11.849 | 14.857 ±3.884 | 6.266 ±12.992 | 7.487 ±12.499 | -3.946 ±4.744 | 1.674 ±6.948 |
| | 6 | 10.016 ±9.847 | -1.836 ±3.545 | 11.016 ±11.884 | 13.063 ±11.705 | 15.903 ±4.963 | 14.745 ±10.973 | 13.074 ±11.386 | -1.877 ±6.103 | 2.735 ±7.084 |
| LIC-TCM | 00025 | 0.766 ±2.535 | 1.678 ±1.992 | -1.869 ±1.975 | -1.866 ±2.123 | 24.702 ±3.313 | -1.785 ±2.099 | -1.957 ±2.144 | -2.207 ±2.125 | -0.885 ±4.961 |
| | 0013 | 4.222 ±2.057 | -1.254 ±1.884 | -2.600 ±2.521 | -0.936 ±2.827 | 23.242 ±2.639 | -0.531 ±2.864 | -1.742 ±2.754 | -3.178 ±2.623 | 0.235 ±4.637 |
| | 005 | 2.927 ±1.723 | -2.666 ±2.048 | -3.465 ±1.789 | -1.411 ±2.734 | 21.533 ±2.264 | -1.390 ±2.653 | -2.222 ±2.577 | -4.662 ±2.423 | 0.808 ±4.886 |
| HiFiC | 014 | 5.833 ±1.006 | 4.700 ±1.472 | 4.021 ±0.791 | 4.539 ±0.684 | 24.169 ±3.136 | 4.643 ±0.657 | 4.253 ±0.740 | 4.020 ±0.802 | -0.256 ±3.419 |
| | 030 | 4.542 ±0.735 | 1.622 ±1.406 | 2.330 ±0.603 | 2.839 ±0.619 | 20.494 ±2.061 | 2.925 ±0.620 | 2.539 ±0.639 | 2.217 ±0.712 | -1.294 ±3.278 |
| | 045 | 5.643 ±1.030 | 2.660 ±1.415 | 3.627 ±0.812 | 4.129 ±0.853 | 24.123 ±2.146 | 4.233 ±0.845 | 3.828 ±0.820 | 3.515 ±0.941 | 0.258 ±3.261 |
| QRes-VAE | 16 | 0.229 ±1.903 | 2.686 ±1.759 | -1.479 ±1.602 | -1.546 ±1.738 | 28.883 ±4.408 | -1.529 ±1.716 | -1.613 ±1.807 | -1.774 ±1.754 | 1.147 ±5.839 |
| | 2048 | 22.571 ±12.064 | -1.529 ±5.434 | 20.070 ±14.531 | 10.621 ±15.356 | 24.029 ±10.946 | 2.104 ±10.099 | 12.397 ±16.284 | -3.928 ±1.382 | 4.582 ±9.555 |
| | 256 | -2.184 ±2.209 | -2.681 ±1.839 | -6.238 ±2.101 | -6.068 ±2.503 | 25.356 ±2.643 | -5.960 ±2.593 | -6.297 ±2.287 | -6.911 ±1.927 | 2.312 ±5.817 |
| CDC | b00032 | 9.188 ±2.021 | 0.156 ±2.353 | 8.082 ±2.933 | 9.630 ±2.556 | 24.192 ±4.128 | 9.728 ±2.463 | 9.199 ±2.758 | 5.834 ±3.621 | 1.488 ±3.366 |
| | b01024 | 12.833 ±1.471 | 5.417 ±2.200 | 10.801 ±1.288 | 10.975 ±1.222 | 32.072 ±3.332 | 10.848 ±1.216 | 10.832 ±1.201 | 10.372 ±1.189 | 5.392 ±2.528 |
| | b02048 | 17.486 ±1.690 | 8.433 ±2.398 | 16.764 ±2.197 | 16.636 ±2.055 | 35.747 ±3.115 | 16.445 ±1.953 | 16.383 ±2.017 | 16.325 ±2.037 | 9.938 ±2.639 |

Table 8: $\Delta$PSNR (↓) scores for different NIC models and their variants across different adversarial attacks. "Reconstruction Loss" objective is used for all attacks.

| NIC | NIC Var. | $\Delta$SSIM↓ | $\Delta$PSNR↓ | $\Delta$MSSSIM↓ | $\Delta$VMAF↓ | $\delta$SSIM↓ | $\delta$PSNR↓ | $\delta$MSSSIM↓ | $\delta$VMAF↓ | $\Delta$BPP↓ |
|---|---|---|---|---|---|---|---|---|---|---|
| Balle et al.(hyperprior) | 1 | -0.117 | -1.3 | -0.012 | 4.26 | 0.155 | 2.19 | 0.036 | 2.52 | 0.00069 |
| | 2 | -0.117 | -1.74 | -0.013 | 4.42 | 0.154 | 2.56 | 0.032 | 2.51 | 0.0029 |
| | 4 | -0.120 | -2.5 | -0.015 | 3.2 | 0.146 | 3.53 | 0.025 | 1.49 | 0.027 |
| | 6 | -0.113 | -1.72 | -0.012 | 2.55 | **0.133** | 5.1 | **0.019** | 1.01 | 0.15 |
| Balle et al.(factorized) | 1 | -0.112 | 0.0572 | -0.009 | 5.87 | 0.162 | 2.39 | 0.040 | 4.11 | 0.00069 |
| | 2 | -0.113 | -0.74 | -0.012 | 4.94 | 0.161 | 2.75 | 0.036 | 3.84 | 0.0029 |
| | 4 | -0.116 | -1.57 | -0.014 | 3.62 | 0.157 | 3.67 | 0.028 | 2.82 | 0.014 |
| | 6 | -0.116 | -1.79 | -0.013 | 2.67 | 0.145 | 5.13 | 0.022 | 1.96 | 0.075 |
| MBT-2018 | 1 | -0.114 | -2.21 | -0.014 | 3.07 | 0.150 | 2.29 | 0.034 | 2.21 | 0.0031 |
| | 2 | -0.119 | **-2.93** | -0.015 | 2.88 | 0.149 | 2.61 | 0.030 | 1.93 | 0.0066 |
| | 4 | -0.113 | -1.92 | -0.012 | 3.13 | 0.146 | 3.77 | 0.026 | 1.57 | 0.042 |
| | 6 | -0.106 | -1.44 | -0.010 | 1.98 | 0.139 | 5.39 | 0.020 | **0.715** | 0.15 |
| MBT-2018(mean) | 1 | -0.114 | -0.254 | -0.011 | 8.54 | 0.156 | 2.26 | 0.037 | 2.14 | -0.00024 |
| | 2 | -0.115 | -0.567 | -0.011 | 8.26 | 0.157 | 2.64 | 0.034 | 2.09 | 7.1e-05 |
| | 4 | -0.119 | -1.49 | -0.014 | 5.01 | 0.146 | 3.54 | 0.025 | 1.06 | 0.024 |
| | 6 | -0.094 | 0.778 | -0.006 | 3.91 | 0.145 | 5.73 | 0.023 | 1.33 | 0.65 |
| JPEG-AI v7.1, BOP | b0002 | -0.095 | 3.56 | -0.006 | 20.6 | 0.186 | 2.42 | 0.064 | 4.86 | 0.015 |
| | b0012 | -0.102 | 2.47 | -0.005 | 13.8 | 0.207 | 3.94 | 0.059 | 6.12 | 0.089 |
| | b0075 | -0.116 | 0.307 | -0.013 | 3.66 | 0.203 | 5.6 | 0.035 | 1.17 | 0.19 |
| | b05 | -0.115 | 0.825 | -0.010 | **0.821** | 0.203 | 7.69 | 0.026 | 1.13 | 0.46 |
| Cheng et al.(attn) | 1 | -0.112 | -1.12 | -0.012 | 4.45 | 0.154 | 2.32 | 0.035 | 2.53 | 0.0012 |
| | 2 | -0.112 | -0.969 | -0.013 | 4.14 | 0.154 | 2.72 | 0.032 | 2.43 | 0.0044 |
| | 4 | -0.110 | -0.505 | -0.012 | 3.91 | 0.149 | 3.9 | 0.026 | 1.92 | 0.041 |
| | 6 | -0.090 | 1.23 | -0.006 | 3.63 | 0.155 | 6.04 | 0.027 | 2.83 | 0.14 |
| ELIC | 0004 | -0.085 | 3.61 | 0.006 | 17.2 | 0.156 | **1.97** | 0.048 | 3.48 | -0.0018 |
| | 0016 | -0.089 | 3.51 | -0.004 | 14.7 | 0.156 | 2.5 | 0.038 | 3.34 | -0.0014 |
| | 0450 | -0.107 | 0.694 | -0.012 | 3.93 | 0.141 | 5.64 | 0.021 | 1.64 | 0.074 |
| LIC-TCM | 00025 | -0.087 | 3.36 | -0.005 | 13.9 | 0.156 | 2.8 | 0.036 | 3.13 | -0.00031 |
| | 0013 | -0.083 | 3.29 | -0.007 | 8.53 | 0.153 | 4.58 | 0.028 | 2.98 | 0.043 |
| | 005 | -0.081 | 2.28 | -0.007 | 4.43 | 0.153 | 6.52 | 0.024 | 2.31 | 0.13 |
| Cheng et al.(anchor) | 1 | -0.110 | -0.678 | -0.012 | 3.9 | 0.154 | 2.36 | 0.035 | 2.37 | 0.0025 |
| | 2 | -0.110 | -0.517 | -0.012 | 3.6 | 0.152 | 2.74 | 0.032 | 2.17 | 0.0081 |
| | 4 | -0.048 | 4.52 | 0.051 | 18 | 0.174 | 7.11 | 0.085 | 16.1 | 0.16 |
| | 6 | -0.008 | 8.2 | 0.084 | 26.7 | 0.186 | 11.1 | 0.110 | 26.2 | 0.2 |
| JPEG-AI v7.1, HOP | b0002 | -0.099 | 3.33 | -0.015 | 16.9 | 0.185 | 2.32 | 0.061 | 2.51 | 0.0061 |
| | b0012 | **-0.125** | 1.39 | **-0.020** | 8.56 | 0.209 | 3.81 | 0.051 | 1.63 | 0.03 |
| | b0075 | -0.111 | 1.03 | -0.011 | 3.78 | 0.223 | 6.23 | 0.044 | 1.35 | 0.15 |
| | b05 | 0.071 | 5.55 | 0.035 | 1.4 | 0.302 | 10.3 | 0.070 | 1.16 | 0.55 |
| QRes-VAE | 16 | -0.084 | 4.51 | -0.002 | 17.4 | 0.147 | 2.46 | 0.031 | 1.67 | -0.00022 |
| | 2048 | 0.183 | 10 | 0.250 | 24.7 | 0.337 | 17.4 | 0.270 | 26.6 | 16 |
| | 256 | -0.108 | 0.91 | -0.011 | 3.54 | **0.136** | 5.83 | **0.019** | 1.22 | 0.091 |
| HiFiC | 014 | 0.065 | 6.78 | 0.048 | 19.6 | 0.250 | 3.1 | 0.088 | 6.03 | 0.0072 |
| | 030 | 0.024 | 4.84 | 0.020 | 10.2 | 0.260 | 4.41 | 0.065 | 6.28 | 0.021 |
| | 045 | 0.050 | 6.57 | 0.034 | 17 | 0.269 | 4.54 | 0.075 | 6.05 | 0.036 |
| CDC | b00032 | 0.111 | 9.08 | 0.080 | 17.7 | 0.271 | 7.33 | 0.107 | 15.7 | 0.093 |
| | b01024 | 0.104 | 13.3 | 0.091 | 43.8 | 0.270 | 7.69 | 0.103 | 28.2 | 0.042 |
| | b02048 | 0.152 | 18.2 | 0.176 | 59 | 0.283 | 9.49 | 0.167 | 27.9 | **-0.0048** |

Table 9: Performance of NIC models across different evaluation metrics averaged across all attacks. "Reconstruction Loss" objective is used as a target for all attacks.

