# OpenReview forum: "NIC-RobustBench: A Comprehensive Open-Source Toolkit for Neural Image Compression and Robustness Analysis"
_ICLR.cc/2026/Conference — Submitted to ICLR 2026_

### Official Review · Reviewer_N61m · 2025-10-31

**Soundness:** 2
**Presentation:** 1
**Contribution:** 2
**Rating:** 4
**Confidence:** 4

**Summary:**

This paper proposes an open source library, NIC-RobustBench, for evaluating the adversarial robustness of state of the art neural image compression (NIC) models. The framework includes a variety of NIC models, white box adversarial attacks, and robustness metrics. Additionally, the paper performs an evaluation of existing models and identifies some interesting trends about what types of models are most vulnerable to attacks. Finally, the paper proposes several defense strategies and analyzes the effects of them in their evaluation.

**Strengths:**

1. The paper includes an open-source package for experimenting with many state-of-the-art NIC codecs.
2. The paper finds interesting trends about types of models and robustness (e.g., larger models are less secure, higher compression rates are the most robust).

**Weaknesses:**

1. The paper has limited novelty and contribution. Specifically, all of the NIC models and attacks are from prior work. Most of the metrics are also borrowed from prior work, with the exception of $\delta$, which the paper does not motivate well. Finally, there is very limited discussions or theoretical insight into the empirical findings.
2. The evaluation datasets are limited and not ideal for image compression tasks. The Cityscapes and NIP 2017 datasets have very low resolution images and Kodak, while having slightly higher resolution, only has 24 images. I would recommend the authors evaluate on CLIC and/or ImageNet.
3. The paper is hard to read in its current state. Specifically, many of the captions are not complete (e.g., what dataset/attack is used in 4a? what dataset in 4b? Where did the images in figure 1 come from?). Many acronyms are presented, but not explained (e.g., what is IQA?, what does VMAF stand for?).  Additionally, the equations include some unclear/inconsistent terminology (e.g., equation 1 is missing an objective (min), equation 2 uses $L(x, x', C(x), C(x'))$, but L is later defined as $L: X \times X \to \mathbb{R}$, and equations 2 and 3 have inconsistent uses of $x'$ and $x+\delta$).

**Questions:**

1. What is the motivation for $\delta$? When should we use $\delta$ vs. $\Delta$?
2. Are there any cases where $\Delta VMAF$ is negative in figure 4b?
2. Do you have any insight into why generative codecs are the most vulnerable? Have you tried comparing generative and discriminative codecs of the same size?

---

> ### Author Response · Authors · 2025-11-26
>
> $\textbf{W1}$. As we mentioned in our answer to reviewer VaBr, our paper states “infrastructure, software libraries, hardware, systems, etc.” as our primary area and presents a software library. We believe that our benchmark, which is the main contribution of this work, will help the community develop more robust and reliable NIC models. Similar benchmarking studies have been accepted at previous ICLR conferences, e.g., [1 - 4].
> See W1 answer to reviewer VaBr for more details on theoretical contributions and novelty. For $\delta_{score}$ motivation, see Q1.
>
> $\textbf{W2}$. Thank you for your suggestion, we will add them to the revised version of the paper.
>
> $\textbf{W3}$. We apologise for misunderstanding. We have fixed these issues and will further enhance the text and increase its clarity. The new version will be in the revised paper.
>
> $\textbf{Q1}$ $\delta_{score}$ motivation: Standard metrics (PSNR and SSIM) evaluate the absolute image quality, but to assess reliability, it is important to understand the relative degradation of compression quality when encoding the attacked image rather than the original one. As shown in Appendix A.5 (Figure 13), there is a correlation between δ and Δ of Spearman correlation = 0.783 (VMAF). This indicates that they capture different aspects of robustness, but they are complementary. Also, $\delta_{score}$ does not consider the visibility of the attack and only measures quality degradation after compression. This makes this metric more interpretable and applicable for robustness assessments in certain tasks (such as downstream tasks where quality is required above a certain threshold and an attack may violate codec requirements). This metric also allows the evaluation of attacks that do not have restrictions on the strength of perturbation.
>
> $\textbf{Q2}$ Figure 4(b) shows the average $\Delta$ VMAF scores, which are all positive. However, during our experiments, negative values were obtained for individual samples but not for the entire dataset. Obtained negative $\Delta$ values were close to zero, indicating that attacks did not work effectively for these cases. Negative $\Delta$ VMAF means that reconstructed images are closer to each other than input images in terms of VMAF, meaning that not only was the attack ineffective, but the attacked codec also filtered out high frequencies introduced by the attack to the original images.
>
> $\textbf{Q3}$ Although our comparison includes models with different parameter counts, we do have a pair where the sizes are nearly identical: QRes-VAE (generative) and ELIC (discriminative), with 34M and 33.8M parameters respectively. Despite having comparable capacity, QRes-VAE is noticeably less robust. We attribute this vulnerability to the way generative codecs rely on a latent representation of the entire image: even small perturbations can shift the latents globally, affecting the reconstructed image as a whole. In contrast, discriminative codecs operate primarily through local, pixel-wise prediction mechanisms. Because they do not depend on a global latent space, perturbations might have a more localized effect and do not propagate through the entire representation, which makes these codecs inherently more robust. Furthermore, previous research [5] that explored the use of NIC models as adversarial defenses for downstream tasks have also demonstrated the greater robustness of discriminative ELIC against generative HiFiC which correlates with our results.
>
> [1] “MMTEB: Massive Multilingual Text Embedding Benchmark” (ICLR 2025):
> Its main novelty is the infrastructure — it offers a collection of tasks, tools and evaluation protocols, which can be used for further theoretical analysis.
> [2] “BatteryML: An Open-source Platform for Machine Learning on Battery Degradation.” (ICLR 2024, "infrastructure, software libraries, hardware, etc." track). BatteryML is an open source software with standardized pipelines for end-to-end machine learning in battery life prediction tasks. The contribution is mostly engineering (framework), rather than new methods or theoretical analysis.​
> [3] "VFLAIR: A Research Library and Benchmark for Vertical Federated Learning" (ICLR 2024, "infrastructure, software libraries, hardware, etc." track). VFLAIR  is an extensible and lightweight VFL framework that integrates existing attacks and defenses for the specific domain.
> [4] “SEAL: A Framework for Systematic Evaluation of Real-World Super-Resolution”(ICLR 2024, "infrastructure, software libraries, hardware, etc." track). The paper presents a framework for systematic evaluation of real-SR. It uses the framework to benchmark existing methods with different protocols.
> [5] Räber S. et al. Human Aligned Compression for Robust Models //arXiv preprint arXiv:2504.12255. – 2025.

---

> ### Author Response · Authors · 2025-12-04
>
> $\textbf{W b)}$.We have added NES and Square attack to our library and evaluated them on the Kodak dataset. The results are shown in Figure 17. To the best of our knowledge, there are no specialized black-box adversarial attacks for the NIC domain. Therefore, the efficiency of the black-box attacks adopted from image classification is significantly lower than that of white-box attacks, justifying the focus of our library on the white-box scenario.
>
> $\textbf{W c)}$. We have added the results on the ImageNet and Clic datasets in Figure 16. The results are rather similar, demonstrating the generalizability of our findings across the chosen datasets.
>
> $\textbf{W d)}$. We have added two new defences: DISCO [5] (purification defence) and MPRNet [6] (denoising defence), originally proposed for the classification domain. The results in Figure 8 show that they are efficient in reducing $\delta VMAF$, therefore those approaches can be efficient as defence for NIC models.

---

### Official Review · Reviewer_VaBr · 2025-10-31

**Soundness:** 3
**Presentation:** 3
**Contribution:** 3
**Rating:** 4
**Confidence:** 4

**Summary:**

This paper introduces NIC-RobustBench, a new open-source library and its associated benchmark for evaluating the adversarial robustness of neural image compression (NIC) methods. The framework supports various adversarial attacks and defense strategies for image compression tasks, enabling a systematic assessment of NIC models' performance in complex adversarial settings. The paper also conducts experiments on several NIC models, examining both attacks and defenses, providing valuable insights into the robustness of NIC models.

**Strengths:**

1. This paper provides a comprehensive compilation of adversarial attack methods, defense strategies, and evaluation metrics for NIC methods, offering a valuable resource for future research in this domain.
2. The experimental section is extensive, covering 10 mainstream NIC models, 6 attack methods, 7 defense strategies, and multiple datasets, offering numerous case studies.
3. The modular design in the framework diagram reflects excellent engineering, facilitating reproducibility and ensuring extensibility for future work.

**Weaknesses:**

1. The paper focuses more on the practical tool aspect and lacks theoretical discussions, particularly regarding the relationship between adversarial robustness and compression mechanisms.
2. The attacks and defense strategies presented in the paper are derived from existing research; the paper mainly integrates these methods rather than proposing novel algorithms, lacking a degree of innovation.

**Questions:**

1. While the paper provides a robust experimental framework, it lacks theoretical analysis. Could the authors further explain why certain NIC architectures, such as generative models, are more susceptible to attacks?
2. Most of the attacks and defense strategies in NIC-RobustBench are based on existing research. Did the authors propose any specific improvements or novel modifications to these methods within the framework?
3. The experimental section is comprehensive, but the analysis of the results is rather superficial. Could the authors provide deeper explanations of experimental phenomena, such as whether an increase in model scale influences the emphasis on image reconstruction, thereby affecting robustness?
4. Did the authors perform experiments to evaluate the impact of adversarial defense methods on RD (Rate-Distortion) performance?
5. We acknowledge the contributions of your work to the field, but it may not fully align with the criteria for ICLR 2026 Main Track. We thus wonder if you would consider submitting it to a more suitable track？

---

> ### Author Response · Authors · 2025-11-26
>
> We appreciate the reviewer for the detailed feedback on our work and would like to point out that the primary area of our paper is “infrastructure, software libraries, hardware, systems, etc.”according to the ICLR 2026 Call for Papers. That may have led to misunderstandings of the proposed contributions.
>
> $\textbf{W1}$. The paper presents the main contribution of the paper as a software library， which is one of the official subject areas of ICLR (“infrastructure, software libraries, hardware, systems, etc.”). Therefore, most experiments are empirical to demonstrate the efficiency and diversity of the experiments that can be performed with the presented framework. Examples of similar benchmark/software libraries papers published on ICLR: [1 - 4]. These works focus on infrastructure, replicability, and providing a unified platform, rather than proposing new algorithms or theories. Their acceptance at ICLR further demonstrates the importance and relevance of these benchmark and library papers within leading research communities.
>
> Some experiments in our work include theoretical analysis (eg. Section A. 3, experiment with NIC model families). We recognize the importance of the theoretical analysis of the NIC robustness, but we consider extensive analysis as a future work. Moreover, this can be significantly simplified with the use of our software.
>
> $\textbf{W2}$. Our paper presents a new benchmarking methodology, which was not proposed before, and combines existing methods implementation in a new, universal and easily scalable benchmarking software. Moreover, our framework covers novel attack strategies by introducing new attack objectives that were not explored in previous work. Most existing research on NIC robustness to quality-degrading attacks only  considered two attack objective formulations: Reconstr. Loss and FTDA Loss (Fig. 2). We have extended the set of possible target functions with 4 additional variants, and our experiments allow us to compare them and analyze their impact on various attack designs, NICs and defenses. We also present a novel $\delta_{score}$ metric that is described in the answer Q1 to reviewer N61m.
>
> $\textbf{Q1}$. Section 6.1 addresses this issue directly. Codecs that employ generative priors in their design, namely CDC, HiFiC and QRes-VAE demonstrate high $\Delta$ and $\delta$ scores (Fig. 5), which means poor adversarial robustness. We note that this effect might be attributed to the larger size of these models compared to other codecs, Section 6.1 demonstrates the correlation of model size with vulnerability (Spearman Corr. 0.724, $p < 10^{−8}$), this correlation is also demonstrated in Figure 7. This finding also aligns well with previous research in other domains [5]. We will further clarify this connection in the revised paper.
>
> $\textbf{Q2}$. See W2 for the answer.
>
> $\textbf{Q3}$. Regarding the question about model scale and reconstruction emphasis: our current results suggest a systematic relationship. As we have shown in Fig. 7, we report a significant positive Spearman correlation ($\rho=0.724$) between the total parameter count of an NIC model and the average attack efficiency ($\Delta/\delta$ scores). Larger models generally achieve better rate–distortion performance on clean images (Fig. 6) by allocating more capacity and bit budget to fine, high-frequency details. However, this also means that the models may have greater number of parameters than required for stable performance, therefore exhibiting lower generalization and robustness.
>
>  In contrast, smaller or more strongly compressed variants (low-BPP settings within the same family) tend to prioritize coarse structure and effectively act as low-pass filters, which suppress a portion of the adversarial signal and thus exhibit higher robustness.
>
> This trend is particularly visible when comparing generative, high-capacity codecs (CDC, HiFiC, QRes-VAE) to lighter, more “classical” architectures (e.g., Ballé factorized/hyperprior, MBT-2018). The former families achieve high perceptual quality but consistently show larger $\Delta$ and $\delta$ scores across attacks and datasets, whereas compact models and low-BPP presets are among the most stable in Fig. 5 and Fig. 7. Within a single model family, the variants with higher compression ratios (lower BPP, fewer effective parameters) are also the most robust, which supports the interpretation that both capacity and reconstruction sharpness contribute to vulnerability.
>
> This analysis is partly already presented in the paper, but we will add the more detailed explanation above to the revised version until the end of rebuttal.

---

> ### Author Response · Authors · 2025-11-26
>
> $\textbf{Q4}$. Yes, we have performed similar experiments on BPP and will add them to the revised version until the end of rebuttal.
>
> $\textbf{Q5}$. See W1 for the answer regarding the primary area chosen and the examples of the accepted ICLR papers with similar contributions in the field of software libraries.
>
> [1] “MMTEB: Massive Multilingual Text Embedding Benchmark” (ICLR 2025):
> Its main novelty is the infrastructure — it offers a collection of tasks, tools and evaluation protocols, which can be used for further theoretical analysis.
> [2] “BatteryML: An Open-source Platform for Machine Learning on Battery Degradation.” (ICLR 2024, "infrastructure, software libraries, hardware, etc." track). BatteryML is an open source software with standardized pipelines for end-to-end machine learning in battery life prediction tasks. The contribution is mostly engineering (framework), rather than new methods or theoretical analysis.​
> [3] "VFLAIR: A Research Library and Benchmark for Vertical Federated Learning" (ICLR 2024, "infrastructure, software libraries, hardware, etc." track). VFLAIR  is an extensible and lightweight VFL framework that integrates existing attacks and defenses for the specific domain.
> [4] “SEAL: A Framework for Systematic Evaluation of Real-World Super-Resolution”(ICLR 2024, "infrastructure, software libraries, hardware, etc." track). The paper presents a framework for systematic evaluation of real-SR. It uses the framework to benchmark existing methods with different protocols.
> [5] Zhu, Zhenyu, et al. "Robustness in deep learning: The good (width), the bad (depth), and the ugly (initialization)." Advances in neural information processing systems 35 (2022): 36094-36107.

---

> > ### Author Response · Authors · 2025-12-04
> >
> > $\textbf{Q4}$. We have added the results for the BPP loss in Fig. 15. The results demonstrate that most defences alter the original curve (without defence) on clean images. Most defences improve SSIM between the input and reconstructed images for the attacked images, demonstrating the efficiency of the defences. The most efficient defenses are Rotate, Self-ensemble, and DiffPure.

---

### Official Review · Reviewer_TW3p · 2025-10-31

**Soundness:** 3
**Presentation:** 2
**Contribution:** 3
**Rating:** 4
**Confidence:** 3

**Summary:**

This paper presents NIC-RobustBench, an open-source, modular benchmark/framework for evaluating the adversarial robustness of neural image compression (NIC) models. It unifies a large set of NIC codecs, six white-box attacks with multiple objectives, and seven defenses and provides a standard pipeline to measure both rate–distortion and robustness, plus impact on downstream CV tasks. The authors then run a broad study over 10+ NIC models and show generative/large codecs are more vulnerable, attack objective matters a lot, and some simple reversible defenses stabilize NICs.

**Strengths:**

a) Compared with existing NIC libraries that mostly focus on RD (CompressAI, TFC, NeuralCompression), this is the first one that systematically bakes in both attacks and defenses, and it clearly has the widest model set so far.
﻿
b) Many “benchmark” papers stop at attacks; this one also implements reversible/geometric defenses and diffusion-style purification (DiffPure), so it supports end-to-end studies.

**Weaknesses:**

a) Novelty is mainly engineering/integration. The paper stands on putting things together, not on a new attack/defense or a new robustness metric. That’s OK for a benchmark, but then it must be very explicit in positioning vs. CompressAI (Bégain et al. 2020), TFC (Ballé et al. 2024), NeuralCompression (Muckley et al. 2021).
﻿
b) Only white-box attacks. The authors argue black-box is costly, but for a benchmark that wants to be comprehensive, skipping black-box / transfer attacks is a real limitation.
﻿
c) Evaluation datasets are small/narrow. KODAK (24 imgs), Cityscapes slice, and NIPS 2017 (1000 imgs) are standard but small; given that NIC is now used in high-res and domain-specific settings, this somewhat weakens the “comprehensive” claim.
﻿
d) Defense set is mostly classical transforms. Apart from DiffPure, the seven defenses are flip/roll/rotate/color/ensembles. This is fine as a baseline, but there is a long line of adversarial purification/denoising/consistency defenses that could be referenced.

e) The image is not good (e.g., Fig 2 and Fig 3). Firstly, no vector graphics were used, and secondly, some fonts are too small

**Questions:**

a) How do you envision adding a “slow but realistic” black-box track (e.g., per-epoch evaluation, cached queries) without breaking the CI-friendly design?
﻿
b) How to evaluate “cost vs. robustness” profile in the benchmark so users can compare defenses at a fixed latency/memory budget?
﻿
c) Some of your findings (larger ≈ less robust; generative ≈ less robust) look quite strong. Do you think these will still hold on higher-res, more diverse datasets, or are they partly an artifact of the current three datasets?

---

> ### Author Response · Authors · 2025-11-26
>
> $\textbf{W a)}$ Yes, our main goal is to make robustness evaluation for NIC standardized and reproducible, which has not been done before and is highly important, since more NIC models are being standardized. Our primary contributions are not only putting NIC models together, but also:
> (i) We introduce a unified family of attack objectives for NIC (Eqs. (2)–(3)) that covers pixel, FR-IQA, and bitrate-oriented losses, which enables a systematic study of how different perturbations affect NIC robustness.
> (ii) We define $\Delta_{score}$ and $\delta_{score}$ (Eqs. (5)–(6)) as generic robustness metrics that extend $\Delta_{PSNR}$ used in previous research on NIC robustness to arbitrary FR-IQA models and distinguish perturbation strength from reconstruction degradation.
> (iii) We adapt several transferable attacks and purification-style defenses to the compression setting and implement them across 10 NIC types and 47 variants.
> Table 1 compares our benchmarks to existing works. Previous benchmarks, including CompressAI and others primarily target rate-distortion training and evaluation in clean, non-adversarial setting, while NIC-RobustBench specifically targets robustness by providing attacks, objectives, defenses, and protocols that these libraries do not currently include. NIC-RobustBench also exceeds all previous works in terms of the number of evaluated models and their variants. We will clarify our contributions and distinctions with other works and in the revised version of the paper.
>
> $\textbf{W b)}$ We agree that, for a fully comprehensive comparison, black-box attacks should also be considered in addition to white-box ones. However, to the best of our knowledge, no successful query-based black-box attacks have yet been proposed specifically for NIC models.
>
> As an intermediate step toward this goal, our current benchmark already evaluates the transferability of attacks in Appendix A.1, which is one of the forms of black-box attacks. Our setup explicitly measures transferability in a black-box evaluation setting, even though the original attack is white-box on the source model.
>
> We agree that adding query-based black-box attacks would further strengthen the benchmark, but this is out of the scope of our current paper goal. Because such attacks are computationally very expensive in the NIC setting, we can only afford a limited number of runs during the rebuttal period. At the end of this response, we therefore provide preliminary results for Square Attack [1] and NES [2], and we will expand these experiments in the final version of the paper, upon acceptance.
>
> $\textbf{W c)}$ First, we would like to clarify that the KODAK dataset is the only one that is currently widely used for NIC evaluation. We additionally added NIPS dataset to support our results in KODAK, but in general, the total amount of images processed throughout our framework scales rapidly with the amount of adversarial attacks, defenses and NICs being applied to clear images. For example, applying 6 attack algorithms with 6 adversarial objectives to target 9 NICs (more accurately, >40 variants with considering different bitrates) will result in >324 images produced from the single clear example (>1400 considering different bitrates). Overall, our calculations required approximately 25000 GPU hours. To further expand the domain of the test setting, we will also include results on subsamples of ImageNet and CLIC databases in the revised paper.
>
> We will include the results on subsample of ImageNet [3] and CLIC [4] in the revised paper variant.
>
> $\textbf{W d)}$ Our purpose was to create a framework that is convenient to scale for new methods (codecs, attacks and defences) of different types. In addition to simple defences, we have added 2 neural-network-based adversarial defenses, DISCO [5] (purification defence) and MPRNet [6] (denoising defence), originally proposed for classification domain and will include the results in the revised version of the paper. To our knowledge, currently, no more complex defences for NIC have been proposed, since this area has recently started to develop. We believe that using our tool, the developers and researchers will be able to easily test the performance of their new defences for NIC models.
>
> $\textbf{W e)}$ To improve clarity of the figures, we divided Figure 2 (a), (b) into separate entries with larger fonts and images (Fig. 2 and 3 in the revised paper). Furthermore, we increased fontsizes in Figure 4. We also note that all figures in the paper are provided in pdf format: all schematics use vector graphics, and text can be selected and copied from the figures.

---

> ### Author Response · Authors · 2025-11-26
>
> $\textbf{Q a)}$ 1) Our CI can host black box scenarios, in revised paper we will add several BB-attacks which can further be modified for real-life cases.
> 2) However, in the area of NIC, promising models must be standardized for wide usage, which means that their code and weights should be open (e.g., to be used on end-users' devices). Thus in this area, the use of white-box attacks might be more prevalent and important than black-box setting.
>
> $\textbf{Q b)}$ Our pipeline features the yaml configs that can be used for changing the parameters of the adversarial attacks and tuning the speed of adversarial attacks. The defences are applied during adversarial attacks like in the previous works, so the desired latency can be achieved by the choice of the attack parameters and desired memory can be achieved by measuring the memory consumption of the NIC model used.
>
> $\textbf{Q c)}$ We will include the experiments on images with higher resolution in the revised version of the paper. However the results on lower resolution datasets in Figure 4 a) demonstrate similar results for all datasets in terms of size and architecture type.
>
> [1] Andriushchenko, Maksym, et al. "Square attack: a query-efficient black-box adversarial attack via random search." European conference on computer vision, 2020.
> [2] Ilyas, Andrew, et al. "Black-box adversarial attacks with limited queries and information." International conference on machine learning, 2018.
> [3] Deng et al. "Imagenet: A large-scale hierarchical image database," IEEE conference on computer vision and pattern recognition, 2009.
> [4] 7th Challenge on Learned Image Compression, https://clic2025.compression.cc/
> [5] Ho, Chih-Hui, and Nuno Vasconcelos. "Disco: Adversarial defense with local implicit functions." Advances in neural information processing systems 35 (2022): 23818-23837.
>
> [6] Mehri, Armin, Parichehr B. Ardakani, and Angel D. Sappa. "MPRNet: Multi-path residual network for lightweight image super resolution." Proceedings of the IEEE/CVF winter conference on applications of computer vision. 2021.

---

> > ### Author Response · Authors · 2025-12-04
> >
> > $\textbf{W2}$. See W c) answer for reviewer N61m.

---

### Author Response · Authors · 2025-12-04
**The summary of the rebuttle**

Dear AC!

Due to the reduced time for communication with reviewers, we would like to summarize the questions raised and the revisions and clarifications we made to improve our paper.

Revisions that address concerns of Reviewer TW3p included text polishing to improve paper positioning, enhancing figures presented in the paper (Figures 2, 3, 4) and extending the available methods with black-box attacks (Figure 17), new datasets (Figure 16) and more defences (Figure 8). Therefore, we have significantly expanded the diversity of approaches in our library.

Most of the Reviewer VaBr's concerns relate to the misunderstanding of the primary area to which our paper is submitted. The primary area of our paper is “infrastructure, software libraries, hardware, systems, etc.” and the work is concentrated on a library that can be used for benchmarking. To address the reviewer's request, we have expanded the library with additional adversarial defense methods and conducted new RD-performance defense evaluations (Figure 15).

Reviewer N61m’s concerns are addressed through evaluation of new datasets (Figure 16) and by improving paper clarity (Figures 2, 3, 4). We have added explanations of the $\delta$ metrics, addressed the points raised in W3, and provided insights into the vulnerability of the generative codecs.

Thank you for taking time to review our submission. We hope we have addressed all reviewers’ concerns about our paper and that it will lead to a favorable decision.

---

### Meta-Review · Area_Chair_FH9D · 2026-01-06

**Summary:**

The three reviewers’ comments are largely consistent and predominantly negative. They raise concerns regarding:
(1) limited novelty—the proposed framework primarily assembles existing neural image compression (NIC) models, attack methods, and defense techniques drawn from prior work;
(2) the small scale of the evaluation datasets; and
(3) the absence of black-box attack settings.

In addition, the reviewers raise several questions concerning technical details and presentation clarity.

AC comments. The stated goal of a robust NIC benchmark framework is to provide meaningful tools and metrics for evaluating the adversarial robustness of different NIC methods. As such, a fundamental question is how the benchmark framework itself is defined, justified, and evaluated. This question is central to any credible benchmark study. Unfortunately, the paper does not address this issue.

**Reviewer Concerns:**

Most concerns are still outstanding.

**Reviewer Scores:**

The reviewers would have likely maintained their respective scores.

---

### Decision · Program_Chairs · 2026-01-26

Reject